# GRADIENT INVERSION TRANSCRIPT: A GENERATIVE MODEL TO RECONSTRUCT TRAINING DATA BY GRADIENT LEAKAGE

## ABSTRACT

We propose Gradient Inversion Transcript (GIT), a generic approach for reconstructing training data from gradient leakage in distributed learning using a generative model. Unlike traditional gradient matching techniques, GIT only requires the model architecture information, without access to the model's parameters, making it more applicable to real-world distributed learning settings. Additionally, GIT operates offline without intensive gradient requests or online optimization. Compared to existing generative methods, GIT adaptively constructs a generative network, with an architecture specifically tailored to the structure of the distributed learning model. Our extensive experiments demonstrate that GIT significantly improves reconstruction accuracy, especially in the case of deep models. In summary, we offer a more effective and theoretically grounded strategy for exploiting vulnerabilities of gradient leakage in distributed learning, advancing the understanding of privacy risks in collaborative learning environments.

## 1 INTRODUCTION

In distributed learning, each client trains its model on local data and shares the gradients with a central server, which aggregates them to update the global model (Jochems et al., 2016; McMahan et al., 2017; Yang et al., 2019). Gradient sharing is also common in federated learning (Huang et al., 2021), but unlike distributed learning, which involves a more centrally coordinated distribution of data across nodes, federated learning (FL) focuses on preserving client privacy by ensuring that data remains localized. While these methods are effective in improving model performance and training efficiency without directly exposing the client's data to public, recent research has shown that sharing gradients can still lead to sensitive information leakage, as attackers may exploit the shared gradients to reconstruct the original training data used by the individual client (Phong et al., 2017; Zhu et al., 2019; Zhao et al., 2020), posing significant privacy risks in real-world distributed learning systems.

There is a considerable amount of work proposed to reconstruct the training data from its gradient (Phong et al., 2017; Zhu et al., 2019; Geiping et al., 2020; Wang et al., 2020; Zhu & Blaschko, 2020; Wu et al., 2023; Pan et al., 2020), based on varying levels of model access. These works can generally be divided into two major categories: gradient matching, which optimizes reconstructed data to align its gradient with the leaked one, and generative methods, which train generative models to map the leaked gradient to the corresponding training data. Gradient matching methods typically need repeated requests for gradients from the model under attack (Zhu et al., 2019; Wei et al., 2020; Geiping et al., 2020; Wang et al., 2020) or full access to the model parameters (Zhu & Blaschko, 2020; Wang et al., 2023), which are usually not satisfied in practice.

We focus on generative methods in this work, which train a generative model called the "threat model" using several input-gradient pairs. The architectures of the threat model are usually predefined in existing methods. That is to say, the architectures of the threat model, such as a multi-layer perception (MLP) (Rosenblatt, 1958) or a UNet Ronneberger et al. (2015), are used irrespective of the model under attack. By contrast, we introduce **Gradient Inversion Transcript (GIT)** in this work to adaptively choose the architecture of the threat model to improve its effectiveness. It is a framework generally applicable to models of different architecture under attack.

**Problem Settings** In this work, we consider a practical distributed learning scenario in which an attacker is able to gain and store the gradient updates sent by each local client but does not have direct access to the clients' raw data or labels. Additionally, the attacker is not able to interact with the central server's global model, meaning the global parameters remain unknown. The attacker also cannot request gradient returns from the global model or modify its architecture to enhance the attack. This setting reflects a more realistic threat model where attackers rely solely on gradient information to attempt data reconstruction.

**Assumptions** Reconstruction by gradient matching has two main assumptions: (1) attackers know private label (Zhu et al., 2019; Wei et al., 2020) or at least label distribution in a data batch (Zhao et al., 2020; Yin et al., 2021; Ma et al., 2023). (2) attackers have access to the back propagation process of the FL model, i.e., attackers are able to obtain returned gradients when they input data (Zhu et al., 2019; Wei et al., 2020; Wang et al., 2020), or global model parameters Zhu & Blaschko (2020). In our settings, similar to prior works that employ generative approaches (Wu et al., 2023; Pan et al., 2020; Huang et al., 2021), we **do not** rely on the above assumptions. Instead, we assume that attackers have access to multiple input-gradient pairs. This setting is more practical, as labels are not shared in distributed learning, and it is challenging for attackers to gain access to the back propagation process.

Our main contributions are as follows:

- We propose a theory-driven training data reconstruction scheme using a generative approach. This method relies solely on gradient information, without requiring access to the backpropagation process or the global model's parameters, as was necessary in previous work. We systematically compare the differences between gradient matching and generative methods, along with their respective attack performance.

- We introduce a new generative model designed based on theoretical derivations. Instead of using a fixed architecture, our generative model is tailored to the structure of the model under attack. Unlike previous empirical approaches, our method is theoretically grounded, resulting in superior performance.

- Unlike gradient matching, our method is based on offline learning. Once the generative model is trained, it can infer the input data without further training, while gradient matching requires repeated online learning for each data batch and necessitates continuous requests for gradients from the global model.

**Notation and Terminology** The federated learning (FL) model from which gradients are leaked to attackers is referred to as the "leaked model," while the network proposed by attackers to reconstruct the training dataset is referred to as the "threat model." In this work, we use $\mathcal{L}_\theta(\boldsymbol{x}, y)$ to represent the loss objective of an FL model, parameterized by $\theta$, on an input-label pair $(\boldsymbol{x}, y)$. The model's weights and batch-averaged gradients are represented by $\mathbf{W}$ and $\nabla \mathbf{W}$, respectively.

## 2 RELATED WORK

Before discussing gradient-based training data reconstruction, it is worth noting that reconstructing datasets using model parameters only is also viable. Methods under this setting require significantly less information than gradient-based methods because they do not need gradient information which is data-dependent. Haim et al. (2022) was the first to reconstruct the training dataset solely based on leaked model parameters by a method grounded in the theoretical analysis from Lyu & Li (2019). Despite using less information, the method is unable to recover high-quality data and fails to achieve pixel-wise accuracy. Consequently, gradient inversion attacks are more widely investigated in the context of the leaked gradients.

**Gradient Matching** Training set reconstruction by gradient matching was initially explored by Phong et al. (2017), which discusses the feasibility of reconstructing training data from shared gradients in distributed learning. Zhu et al. (2019) demonstrated its practicality by proposing a method called Deep Leakage from Gradients (DLG). DLG optimizes a randomly generated dummy input to match the training data by minimizing the distance between the dummy gradients and the leaked ground truth gradients. Building on DLG, Wei et al. (2020) evaluate the impact of different federated learning configurations, such as batch size, on the performance of gradient matching. Geiping

et al. (2020) extend DLG by leveraging only the direction of the gradient and replace the optimizer LBFGS with Adam. Wang et al. (2020) propose a Gaussian-kernel-based cost function to reconstruct training data at any training phase. Zhu & Blaschko (2020) introduce a closed-form recursive procedure to recover data in which all gradients and parameters are exposed to the attacker. Furthermore, Wang et al. (2023) propose a provable gradient inversion attack focusing on reconstructing a batch of data by querying a model with malicious parameter.

**Reconstruction By Generative Models** Unlike reconstruction by gradient matching, the generative approaches train a threat model to generate the reconstructed training data with the leaked gradients as the input. The idea of employing a generative model for training data reconstruction was originally proposed in Wu et al. (2023), which uses a three-layer MLP with fixed hidden size as the generated model. Pan et al. (2020) propose a theoretically grounded method to train generative models, leveraging the presence of exclusively activated neurons. In addition, Huang et al. (2021) demonstrate that generative techniques can exhibit strong performance even when attackers lack access to precise batch norm statistics. Furthermore, pretrained generative models, such as the ones trained on other samples from the training data distribution (Jeon et al., 2021) or public datasets (Li et al., 2022), have also shown the potential to improve the performance of generative training data reconstruction.

The mentioned generative methods above employ a threat model of a fixed architecture regardless of the leaked model, which may not be optimal. In contrast, we introduce a framework that dynamically selects the architecture of the threat model based on the leaked model to enhance performance.

**Challenges of Training Data Reconstruction** One key challenge is to restore the label information, which is the key to reconstructing the training data. Although many methods require the attacker's access to the label information (Zhu et al., 2019; Wei et al., 2020) or label distribution (Zhao et al., 2020; Yin et al., 2021; Ma et al., 2023), several attempts have been made to restore the label information based on the leaked gradients. These methods usually tackle one particular scenario or have additional assumptions, including small batch size (Zhao et al., 2020), no duplicate labels in a mini-batch (Yin et al., 2021), and access to the output probability of each class (Ma et al., 2023). By contrast, a recent work Chen & Vikalo (2024) considers a more realistic scenario, which takes multiple local epochs, heterogeneous data and various optimizers into consideration.

Another key challenge is dealing with large batch sizes. The dimensionality of the leaked gradient is fixed, but a large batch size means more information to reconstruct. Restoring the label information has been shown effective in improving the performance in large batch size regime (Yin et al., 2021). In addition, there are several works (Fowl et al., 2021; Wen et al., 2022; Wang et al., 2023; Hayes et al., 2024) proposed to improve the performance of reconstructing large batch training data under different settings. However, there are still considerable performance gaps between small batch and large batch regimes.

In our framework, we do not assume any access to the label information. In addition, we evaluate our methods against baselines across varying batch sizes. Comprehensive experiments validate the effectiveness of our methods despite these challenges.

## 3 ANALYTIC GRADIENT INVERSION ATTACK

### 3.1 RECONSTRUCTION OF LINEAR MODEL

We first consider an $N$-layer feedforward neural network as follows:

$$\mathcal{L}_\theta(\boldsymbol{x}, y) = \ell(\boldsymbol{z}_N, y) = \ell(\mathbf{W}_N \boldsymbol{a}_{N-1}, y); \; \boldsymbol{a}_i = \sigma_i(\boldsymbol{z}_i), \; \boldsymbol{z}_i = \mathbf{W}_i \boldsymbol{a}_{i-1}, \; i = 1, 2, ..., N-1 \quad (1)$$

Here, we denote the width of the neural network or namely the number of hidden nodes for the $i$-th layer as $\{d_i\}_{i=1}^{N-1}$. The input data batch $\boldsymbol{a}_0 = \boldsymbol{x} \in \mathbb{R}^{B \times d_0}$, where $B$ is the batch size. In addition, we define $\boldsymbol{a}_N = \mathbf{W}_N \boldsymbol{a}_{N-1}$ as the output logit of the model. $\{\mathbf{W}_i \in \mathbb{R}^{d_i \times d_{i-1}}\}_{i=1}^N$ refer to the parameters of $N$ linear layers, including convolutional layers and fully connected layers. $\{\sigma_i\}_{i=1}^{N-1}$ are the nonlinear activation functions of different layers. $\boldsymbol{z}_N = \mathbf{W}_N \boldsymbol{a}_{N-1}$ is the output logit, and $\ell$ is the function calculating the classification error, such as the softmax cross-entropy function. In this context, $\{\boldsymbol{z}_i \in \mathbb{R}^{B \times d_i}\}_{i=1}^{N-1}$ and $\{\boldsymbol{a}_i \in \mathbb{R}^{B \times d_i}\}_{i=1}^{N-1}$ represent the pre-activation and post-activation of intermediate layers, respectively.

We use $\boldsymbol{g}_i = \nabla_{\mathbf{W}_i} \mathcal{L}_\theta(\boldsymbol{x}, y)$ to represent the gradient of each weight matrix. In distributed learning or federated learning, each client reports gradient averaged on their local data batch $S$ with size $B$, i.e., $\bar{\boldsymbol{g}}_i := \frac{1}{B} \Sigma_{b=1}^{B} \nabla_{\mathbf{W}_i} \mathcal{L}_\theta(\boldsymbol{x}^{(b)}, y^{(b)})$, $S = \{(\boldsymbol{x}^{(1)}, y^{(1)}), (\boldsymbol{x}^{(2)}, y^{(2)}), ..., (\boldsymbol{x}^{(B)}, y^{(B)})\}$. Based on back-propagation, we have the following equations according to the chain rule:

$$\boldsymbol{g}_i = \left( \prod_{j=i}^{N-1} \mathbf{W}_{j+1}^T \odot \sigma_j'(\boldsymbol{z}_j) \right) \otimes \frac{\partial \mathcal{L}}{\partial \boldsymbol{z}_N} \otimes \boldsymbol{a}_{i-1}^T, \; i = 1, 2, ..., N \tag{2}$$

Here we define two operators, namely $\otimes$ and $\odot$. $\otimes$ denotes tensor multiplication. $\odot$ denotes broadcast row-wise product. Specifically, we let $\mathbf{W}_{j+1}^T \odot \sigma'(\boldsymbol{z}_j) := V_j \in \mathbb{R}^{B \times d_j \times d_{j+1}}$ where $V_j[i_1, i_2, :] = \sigma_j'(\boldsymbol{z}_j[i_1, i_2]) W_{j+1}^T[i_2, :]$. In addition, $\frac{\partial \mathcal{L}}{\partial \boldsymbol{z}_N}$ is broadcast as a tensor of a shape $B \times d_N \times 1$ and $\boldsymbol{a}_{i-1}^T$ is broadcast as a tensor of a shape $B \times 1 \times d_{i-1}$. Therefore, $\boldsymbol{g}_i \in \mathbb{R}^{B \times d_i \times d_{i-1}}$ is a third-order tensor. This tensor encapsulates the gradient information across the entire batch. In distributed learning or federated learning, we average it along the batch dimension before sharing it with the central server, formally expressed as $\bar{\boldsymbol{g}}_i = \mathbb{E}_b[\boldsymbol{g}_i[b, :, :]]$.

Based on Equation (2), we can approximate the value of $\boldsymbol{a}_{i-1}^T$ as follows:

$$\boldsymbol{a}_{i-1}^T \simeq \left( \frac{\partial \mathcal{L}}{\partial \boldsymbol{z}_N} \right)^+ \otimes \prod_{j=N-1}^{i} \left( \mathbf{W}_{j+1}^T \odot \sigma_j'(\boldsymbol{z}_j) \right)^+ \otimes \boldsymbol{g}_i, \; i = 1, 2, ..., N \tag{3}$$

Here, we use $(\cdot)^+$ to represent the Moore–Penrose inverse of a matrix. For a third-order tensor, $(\cdot)^+$ calculate the Moore-Penrose inverse of each of its subspace via the first dimension. Similar to Equation (2), we broadcast $\frac{\partial \mathcal{L}}{\partial \boldsymbol{z}_N}$, $\boldsymbol{a}_{i-1}^T$ and treat them as third-order tensors. Approximation in (3) still involves the product of a sequence, but we can re-organize (3) to approximate $\boldsymbol{a}_{i-1}$ by $\boldsymbol{a}_i$:

$$\boldsymbol{a}_{i-1}^T \simeq \boldsymbol{a}_i^T \otimes \boldsymbol{g}_{i+1}^+ \otimes (\mathbf{W}_{i+1}^T \odot \sigma_i'(\boldsymbol{z}_i))^+ \otimes \boldsymbol{g}_i, \; i = 1, 2, ..., N-1 \tag{4}$$

Applying (4) iteratively, we can derive a recursive training data reconstruction method, which propagates from $\boldsymbol{a}_N$ to $\boldsymbol{a}_0$ and thereby facilitate the recovery of the original training data.

## 3.2 RECONSTRUCTION OF ACTIVATION FUNCTION

The right hand side of (4) involve the term $\sigma'(\boldsymbol{z}_i)$ which introduces nonlinearity. When applying (4) iteratively, we can estimate the value of $\sigma'(\boldsymbol{z}_i)$ based on $\boldsymbol{a}_i$. Since both $\sigma$ and derivative of $\sigma$ are applied elementwisely, the mapping from $\boldsymbol{a}_i$ to $\sigma'(\boldsymbol{z}_i)$ is also elementwise. Although function $\sigma_i$ may not be an injective function, we demonstrate in Table 1 below that we can uniquely identify $\sigma'(\boldsymbol{z}_i)$ given $\boldsymbol{a}_i$ for the most popular activation functions used in practice.

| Name | ReLU | Leaky ReLU | Sigmoid | Tanh |
|---|---|---|---|---|
| $\boldsymbol{a}_i = \sigma_i(\boldsymbol{z}_i)$ | $\max(0, \boldsymbol{z}_i)$ | $\max(k\boldsymbol{z}_i, \boldsymbol{z}_i)$ | $\frac{1}{1+e^{-\boldsymbol{z}_i}}$ | $\frac{e^{\boldsymbol{z}_i} - e^{-\boldsymbol{z}_i}}{e^{\boldsymbol{z}_i} + e^{-\boldsymbol{z}_i}}$ |
| $\sigma_i'(\boldsymbol{z}_i)$ | $\begin{cases} 1 & \text{if } \boldsymbol{a}_i > 0 \\ 0 & \text{if } \boldsymbol{a}_i = 0 \end{cases}$ | $\begin{cases} 1 & \text{if } \boldsymbol{a}_i > 0 \\ k & \text{if } \boldsymbol{a}_i \leq 0 \end{cases}$ | $\boldsymbol{a}_i(1 - \boldsymbol{a}_i)$ | $1 - \boldsymbol{a}_i^2$ |

Table 1: Mappings from $\boldsymbol{a}_i$ to $\sigma'(\boldsymbol{z}_i)$ for popular activation functions. Operations are elementwise.

In practice, when we are using ReLU as the activation function, $\sigma'(\boldsymbol{z}_i)$ will be a sparse matrix. This may cause numerical instability when we calculate $(\mathbf{W}_{i+1}^T \odot \sigma_i'(\boldsymbol{z}_i))^+$ on the right hand side of (4). In this case, we replace zero elements with a small pre-defined constant $\epsilon$ in $\sigma'(\boldsymbol{z}_i)$.

## 3.3 MORE GENERAL ARCHITECTURE

Equation (1) formulates a feedforward neural network consisting of linear layers and activation functions alternatively. In practice, we may use more complicated architecture to boost performance. For example, skip connections are widely used in deep neural networks: their application in ResNet (He et al., 2016) has proven effective in addressing challenges such as gradient vanishing. Therefore, it is necessary and important to generalize the analyses above to these architectures.

Without the loss of generality, we consider a neural network with one single shortcut connection which links $k$-th layer to $l$-th layer ($k < l$). Specifically, the shortcut connection links the post-activation $\boldsymbol{a}_k$ to the pre-activation $\boldsymbol{z}_l$ with a weight parameter $\mathbf{S} \in \mathbb{R}^{d_k \times d_l}$. Therefore, $\{\boldsymbol{z}_i\}_{i=1}^N$ and $\{\boldsymbol{a}_i\}_{i=1}^N$ are calculated in the same manner except that $\boldsymbol{z}_l = \mathbf{W}_l \boldsymbol{a}_{l-1} + \mathbf{S} \boldsymbol{a}_k$. Based on the back propagation, $\boldsymbol{g}_i$ is calculated in the same way as in Equation (2) when $i > k$. When $i \leq k$, $\boldsymbol{g}_i$ is calculated as follows. For notation simplicity we define $M_j = \mathbf{W}_{j+1}^T \odot \sigma_j'(\boldsymbol{z}_j)$.

$$\boldsymbol{g}_i = \prod_{j=i}^{k-1} M_j \otimes \left( \prod_{j=k}^{l-1} M_j + \mathbf{S} \odot \sigma_k'(\boldsymbol{z}_k) \right) \otimes \prod_{j=l}^{N-1} M_j \otimes \frac{\partial \mathcal{L}}{\partial \boldsymbol{z}_N} \otimes \boldsymbol{a}_{i-1}^T \tag{5}$$

Following a similar analysis to (3) and (4), we can derive an approximation of $\boldsymbol{a}_{i-1}$ using $\boldsymbol{a}_i$. The approximation is the same as (4) except for the case $i = k$. This is because the shortcut connection contributes to the gradient $\boldsymbol{g}_k$ but not $\boldsymbol{g}_{k+1}$: $\boldsymbol{g}_{k+1}$ is calculated based on Equation (2) while $\boldsymbol{g}_k$ is calculated based on Equation (5). In this regard, combining Equation (2) with $i = k + 1$ and Equation (5) with $i = k$, we obtain the following approximation:

$$\boldsymbol{a}_{k-1}^T \simeq \left( \left( \mathbf{W}_{k+1}^T \odot \sigma_k'(\boldsymbol{z}_k) \right) \otimes \boldsymbol{g}_{k+1} \otimes (\boldsymbol{a}_k^T)^+ + (\mathbf{S} \odot \sigma_k'(\boldsymbol{z}_k)) \otimes \boldsymbol{g}_l \otimes (\boldsymbol{a}_{l-1})^+ \right)^+ \otimes \boldsymbol{g}_k \tag{6}$$

Compared to (4), the estimation in (6) incorporates not only $\boldsymbol{g}_k$ and $\boldsymbol{g}_{k+1}$ but also $\boldsymbol{g}_l$ to estimate $\boldsymbol{a}_{k-1}^T$. Since $\boldsymbol{a}_k$ is connected to $\boldsymbol{z}_l$ via skip connection, gradients can flow directly from the $l$-th layer to the $k$-th layer in back propagation. The insight provided by approximation (6) reveals how preceding activations are estimated based on gradients in a general neural network architecture. The reconstruction sequence aligns with the gradient flow during back propagation. In the subsequent section, we delve into the implementation of such reconstruction using a generative model.

## 4 METHODOLOGY: GRADIENT INVERSE TRANSCRIPT

Building upon the principles and assumptions of distributed learning and federated learning as elucidated in Section 1, we train a generative model, denoted as the threat model, utilizing multiple input-gradient pairs $(\boldsymbol{x}, \{\boldsymbol{g}_i\}_{i=1}^N)$. Note that we do not have any knowledge about the leaked model other than its architecture and do not have the access to call back propagation as in DLG Zhu et al. (2019). In addition, we do not have access to the parameters of the leaked model or the label of the training data. Upon completion of training, the threat model utilizes the leaked gradients as input to generate the training data batch as output.

Most existing generative reconstruction methods use fixed architectures (Zhu et al., 2019; Li et al., 2022), such as multi-layer perceptrons (MLP) or UNets. However, these designs are heuristic and may not be the optimal for leaked models of different architectures. Based on the analyses in Section 3, we propose a novel generative reconstruction scheme called Gradient Inverse Transcript (GIT), illustrated in Figure 1. In approximation (4) and (6), all the variables except the gradients $\{\boldsymbol{g}_i\}_{i=1}^N$ are unknown. In this regard, we can represent the unknown variables as the trainable parameters of the generative model. By applying approximation (4) and (6) iteratively, we can build a neural network as the generative model to reconstruct the training data. It is important to note that the architecture of this generative model adapts to the one of the leaked model and is a "translation" of its back propagation as demonstrated in Figure 1. (A more general architecture for networks with skip connections are shown in Appendix D.)

Based on the analyses in Section 3.2, the value of $\{\sigma_i'(\boldsymbol{z}_i)\}_{i=1}^{N-1}$ can be calculated based on the estimated value of $\{\boldsymbol{z}_i\}_{i=1}^{N-1}$. By applying the approximation (4) or (6) iteratively, we can find $\{\mathbf{W}_i\}_{i=1}^N$ are the only unknown variables, so we include these variables as model parameters in the threat model. In addition, we need the value of $\frac{\partial \mathcal{L}}{\partial \boldsymbol{z}_N}$ to estimate the value of $\boldsymbol{a}_{N-1}$ by $\boldsymbol{a}_{N-1} \simeq \left( \frac{\partial \mathcal{L}}{\partial \boldsymbol{z}_N} \right)^+ \otimes \boldsymbol{g}_N$ so that we can iteratively estimate the value of the preceding layers. When the last layer of the neural network has a bias term $\boldsymbol{b}_N$, i.e., $\boldsymbol{a}_N = \mathbf{W}_N \boldsymbol{a}_{N-1} + \boldsymbol{b}_N$, following the idea of Ma et al. (2023), we have $\frac{\partial \mathcal{L}}{\partial \boldsymbol{z}_N} = \frac{\partial \mathcal{L}}{\partial \boldsymbol{b}_N}$. That is to say, we can directly utilize the gradient of the bias term in the last year as $\frac{\partial \mathcal{L}}{\partial \boldsymbol{z}_N}$. When the last layer of the neural network does not have a bias term, we cannot directly obtain $\frac{\partial \mathcal{L}}{\partial \boldsymbol{z}_N}$. In addition, considering that $\frac{\partial \mathcal{L}}{\partial \boldsymbol{z}_N}$ depends on the input

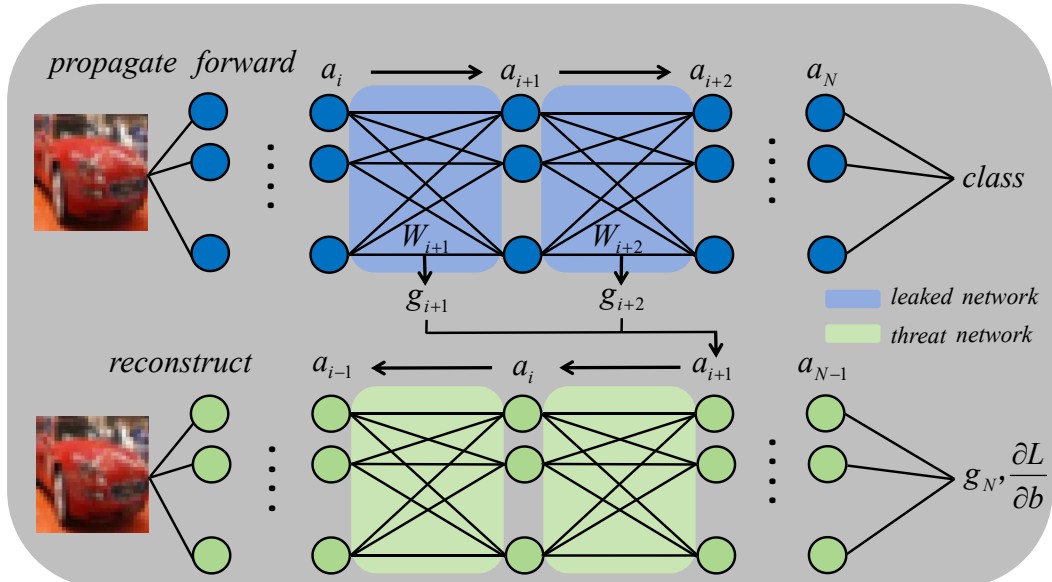

Figure 1: **(Top half)** The leaked model which leaks the gradient to the attackers. **(Bottom half)** The threat model constructed by Inverse Gradient Transcript (GIT) based on the approximation (4). The threat model is a generative model utilizing the leaked gradients to reconstruct the training mini-batch data. In FineGIT mode, we estimate $a_i$ based on the approximation (4) with unknown variables as trainable parameters. In CoarseGIT mode, we use an MLP to estimate $a_i$ with the gradient and activation estimation based on (4) as the input.

data, we cannot treat it as a parameter, either. In this scenario, we introduce a multi-layer perception (MLP) model to concatenate the gradient information $\{g_i\}_{i=1}^N$ and map it to $\frac{\partial \mathcal{L}}{\partial z_N}$. This MLP model is trained jointly with the threat model.

In addition to strictly following the computation in the backward estimation such as the one in (4) and (6) and only including $\{\mathbf{W}_i\}_{i=1}^N$ as the parameters of the threat model, we can also model the inference from $a_{N-1}$ to its preceding layers in a more coarse-grained manner. Specifically, for the $i$-th layer we use a shadow but nonlinear multi-layer perception (MLP) model represented by the function $m_i$ to model the mapping from $a_i$ to $a_{i-1}$. Besides $a_i$, the inputs of this MLP also include the gradient information used to infer $a_i$. That is to say, based on the topology of the neural network, when we use approximation (4), we have $a_{i-1} = m_i(a_i, g_{i+1}, g_i)$; when we use approximation (6), we have $a_{k-1} = m_k(a_k, g_{k+1}, a_{l-1}, g_l, g_k)$. We do not include the Moore-Penrose inverse in the formulation, because we find it may cause numerical instability and the MLP employed here has the capacity to model the inverse operation. Under this coarse setting, the threat model is the composition of these MLP models, which are trained jointly.

Based on the parameterization of the threat model discussed above, we name the corresponding methods Fine-grained Gradient Inverse Transcript (FineGIT) and Coarse-grained Gradient Inverse Transcript (CoarseGIT), respectively. FineGIT is more aligned with the back propagation calculation and has fewer parameters to train, but it lacks flexibility and may suffer from numerical instability. This is because we use the approximated value of $\{a_i\}_{i=1}^{N-1}$ to estimate $\{\sigma_i'(z_i)\}_{i=1}^{N-1}$, which may cause approximation error to propagate. In addition, we need to calculate the Moore-Penrose inverse of the trainable parameters in approximation (4) and (6), which may cause numerical instability, especially in the cases of low-rank matrices. CoarseGIT, on the other hand, is more flexible, stable but has more parameters to train. Our observation in practice indicates that FineGIT is more stable when the leaked model's feature map is smaller and the model's width is narrow.

When training the threat model, we use mean squared error $\|a_0 - \widehat{a}_0\|^2$ as the loss objective function where $a_0 = x$ is the ground truth mini-batch inputs and $\widehat{a}_0$ is the estimation for the input data by the threat model.

We formally present analytic reconstruction procedure of GIT in Algorithm 1 in the appendix.

## 5 EXPERIMENTS

We assess our methods on classification tasks using the CIFAR-10 (Krizhevsky et al., 2009) image dataset. In the realm of distributed learning or federated learning, a central server refines a classification model by aggregating gradients shared by user devices, derived from their individual training data. Our experiment operates under the assumption that user-side local datasets are subsets of CIFAR-10. The attacker, with access to a subset of gradient-input pairs, endeavors to reconstruct the remaining input data using the gradients shared by others. These pairs for training the threat models are sampled from CIFAR-10's training set (unless otherwise specified, in the subsequent experiments, we use one-tenth of the training data, which consists of 5,000 samples), we evaluate the performance of the recontruction methods on CIFAR-10's test set. To quantitatively evaluate model efficacy, we utilize mean squared error (MSE) as the metric for evaluating the performance of training data reconstruction.

We mainly use LeNet (LeCun et al., 1998) and ResNet (He et al., 2016) of various depth as the architecture of the leaked model in our experiments. We use the approximation in (4) for LeNet and the approximation in (6) for ResNet, since ResNet includes shortcut connections. For layers other than linear layers, including pooling layers and batch normalization layers (Ioffe, 2015), as discussed in Section 3.3, we can construct the corresponding architecture of the threat model based on the back propagation through these layers.

**Baselines** We benchmark our methods against two approaches: (1) Deep Leakage from Gradients (DLG) (Zhu et al., 2019), which belongs to the category of gradient matching methods; (2) The generative approach utilizing a fixed MLP architecture (Wu et al., 2023). We select these two as our baselines, because both of them achieve competitive performance in their respective category. For generative methods, we do not use UNet as the fixed architecture, because UNet-based generative models leverage priors from the public data. However, we do not assume any access to the public data by the attacker.

Although the gradient matching methods diverge from our assumptions and configurations studied, we opt to compare with these methods due to its widespread application. Gradient matching techniques necessitate a complete optimization process for each batch data recovery, whereas generative methods need to train a generative model capable of retrieving data from any batch used in its training. That is to say, the major computational overhead for gradient matching methods is the per-batch optimization process during reconstruction, while the major overhead for generative methods is to train a generative model. To ensure a fair comparison, we keep the computational complexity approximately the same for methods of both categories.

### 5.1 RECONSTRUCTION IN VARIOUS BATCH SIZES AND NETWORK ARCHITECTURES

Table 3 compares the performance of our proposed method (GIT) against baselines across different network architectures and batch sizes. The results indicate that the reconstruction is more challenging with a larger batch size and a deeper architecture. Our proposed GIT outperforms baselines in all cases except LeNet with batch size being 1, where DLG performs the best and almost perfectly recover the input data. It is not surprising because DLG can obtain more information from the model through repetitive online requests. However, as the batch size increases, DLG's performance declines significantly, revealing its inability to handle larger batches effectively. GIT, on the other hand, outperforms other methods when the batch size exceeds 1, indicating its ability to recontruct multiple input data at the same time.

Among the generative models, GIT outperforms the baseline that uses a fixed MLP as the threat model in all cases. The results validate the effectiveness of using an adaptive architecture for the threat model as discussed in Section 3. Moreover, we notice the issue of overfitting when training generative models. Specifically, the training loss in MSE for both GIT and MLP models can drop below 0.005 while the test loss demonstrated in Table 3 is significantly larger. We believe the overfitting issue arises from insufficient training data and lack of regularization schemes. We leave mitigating overfitting of generative reconstruction methods as our future works. The first 8

reconstructed images in CIFAR-10 test set are shown in Appendix E.1, illustrating the visual quality corresponding to the first column of Table 3.

Table 2: Quantitative Comparison of GIT with prior works on different networks and batch sizes. We use MLP & UNet to represent the generative method using a fixed MLP & UNet architecture, which shares similar number of parameters to our proposed method. The numbers in the table represent the MSE & PSNR between the reconstructed data and the ground truth on the test set. We use $10000$ samples to train generative models.

| Leaked Model | Method | Metrics | Batch Size = 1 | Batch Size = 2 | Batch Size = 4 |
|---|---|---|---|---|---|
| LeNet (5 layers) | DLG | MSE | **0.0008** | 0.0472 | 0.0975 |
| | | PSNR | **30.97** | 13.26 | 10.11 |
| | MLP | MSE | 0.0241 | 0.0332 | 0.0571 |
| | | PSNR | 16.18 | 14.79 | 12.43 |
| | GIT | MSE | 0.0099 | **0.0122** | **0.0254** |
| | | PSNR | 20.04 | **19.14** | **15.95** |
| | UNet | MSE | 0.0316 | 0.0393 | 0.0435 |
| | | PSNR | 15.00 | 14.06 | 13.62 |
| ResNet (20 layers) | DLG | MSE | 0.1202 | 0.1347 | 0.1365 |
| | | PSNR | 9.20 | 8.71 | 8.65 |
| | MLP | MSE | 0.0354 | 0.0473 | 0.0589 |
| | | PSNR | 14.51 | 13.25 | 12.30 |
| | GIT | MSE | **0.0193** | **0.0246** | **0.0388** |
| | | PSNR | **17.14** | **16.09** | **14.11** |
| | UNet | MSE | 0.0515 | 0.0560 | 0.0619 |
| | | PSNR | 12.88 | 12.52 | 12.02 |

Table 3: Results for different methods with varying batch sizes and network depths.

For datasets with larger resolutions, such as TinyImageNet-200, the MSE for varying batch sizes and network architectures, along with the reconstructed images, are presented in Appendix E.2. The results demonstrate that high-frequency information, including object contours and background details, is effectively recovered. Although the MSE and visual quality are lower compared to CIFAR10 under the same configuration, reconstructing data from higher-resolution images poses a significant challenge. Notably, resolutions larger than CIFAR10 have not been explored in baseline methods MLP and DLG.

## 5.2 RECONSTRUCTION BY NOISY GRADIENTS

Gradient perturbation is a commonly used defense method against gradient leakage (Zhu et al., 2019). As shown in prior work Wu et al. (2023), the generative model demonstrates superior performance over DLG in countering privacy defenses. Our results in the left half of Table 4 validate this conclusion for GIT when encountering gradient perturbation with varying noise variance. In addition, GIT demonstrates better performance than using a fixed MLP model in all cases. We apply Gaussian noise with standard deviation (std) of $0.01$ and $0.1$. DLG is shown to be highly sensitive to the noise added to the gradients, Gaussian noise with a std of $0.01$ is sufficient to prevent DLG from accurately recovering the input image, and noise with a std of $0.1$ will result in reconstructed images being entirely comprised of noise. In contrast, GIT maintains a mean squared error (MSE) of approximately $0.01$ even when the noise std reaches $0.1$, showing minimal susceptibility to noise. In addition, we notice that the sixth recovered image in the validation set shows an inverse trend compared to the other seven images, where the quality improves as the training dataset size decreases. This anomaly could be attributed to the image being an outlier in the CIFAR-10 distribution. We will leave this observation for future work.

Table 4: Comparison of the MSE under gradient perturbation with varying noise variance (left) and varying volumes of training data (right). The batch size is fixed at 1, and the leaked model is LeNet with 5 layers.

| std of noise | DLG | MLP | GIT | Volume | GIT | MLP |
|---|---|---|---|---|---|---|
| None | **0.001** | 0.024 | 0.009 | 1000 | **0.016** | 0.035 |
| 0.01 | 0.105 | 0.024 | **0.009** | 5000 | **0.013** | 0.028 |
| 0.1 | 0.163 | 0.024 | **0.010** | 10000 | **0.009** | 0.024 |

## 5.3 RECONSTRUCTION BY DIFFERENT VOLUMES OF TRAINING DATA

In this section, we evaluate the performance of GIT using varying amounts of training data: $1,000$, $5,000$ and $10,000$ samples. The right half of Table 4 shows impact of training data volume on generative approach. Considering the generative models can achieve almost the perfect performance on the training set, we can conclude that a larger training set can help mitigate overfitting and thus enhance the performance of the model. In addition, GIT is shown to achieve better performance than using a fixed MLP architecture in all cases. The first 8 reconstructed images in CIFAR-10 test set is illustrated in Figure 6. It shows that even with only 1000 input-gradient pairs, GIT is still able to reconstruct reasonable images, indicating that with a small amount of training data, effective recovery is still achievable.

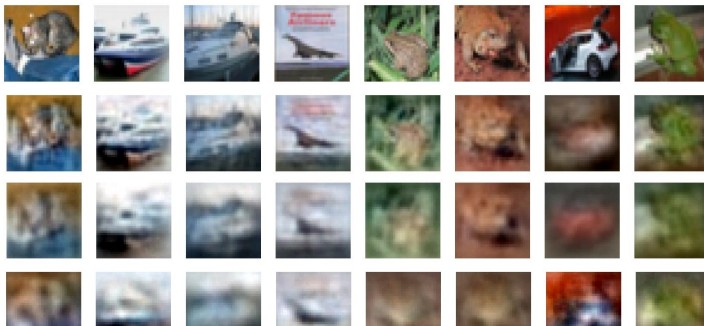

Figure 2: Comparison the first 8 reconstructed images in CIFAR-10 test set using different amount of training data. The leaked model model is LeNet and batch size is 1. (**From top to bottom**) ground truth images, reconstructed images using 10000 samples, reconstructed images using 5000 samples and reconstructed images using 1000 samples

## 5.4 ABLATION STUDIES FOR MODEL COMPLEXITY

Since GIT learns to invert gradients based on the architecture dependent on the model under attack, its model complexity varies for different FL models and differs from the generative approach that employs a fixed architecture. Generally speaking, during inference, the complexity of running GIT is proportional to running the model under attack, because their architectures are related.

To understand this dependence, we conduct ablation studies by varying the depth and width of an MLP. The results are shown in table 5. All ablation studies are performed with a LeNet FL model and a batch size of 1. In the generative approach with a fixed MLP, a three-layer structure is adopted, each with 1000 hidden units. The first ablation study keeps the model depth constant but increases the depth of each layer to match the number of parameters of GIT, allowing us to evaluate the differences between our method and a standard MLP under the same model complexity. The second experiment maintains the hidden size of each MLP layer but increases the model depth to match that of GIT. However, in these configurations, all gradients are flattened and input from the

first layer, rather than fed incrementally through layers as in Coarse GIT. The results show that our proposed GIT model performs better than all configurations of baselines, highlighting its superior performance under fair conditions of equal computational complexity.

Table 5: Ablation Studies for Model Complexity. All ablation studies are performed with a LeNet FL model and a batch size of 1.

| MLP | MLP with Fixed Depth | MLP with Fixed Width | GIT |
|---|---|---|---|
| $0.0241 \pm 0.0003$ | $0.0224 \pm 0.0003$ | $0.0129 \pm 0.0004$ | $\mathbf{0.0099 \pm 0.0001}$ |

### 5.5 ANALYSIS OF THE TREND OF LEARNED WEIGHTS

In this section, a complementary experiment is conducted to measure the L2 distance between the weights of the optimized neural network and those of the leaked neural network. This serves as an additional metric for evaluating the effectiveness of GIT. The experiment is conducted on Fine-GIT, as its parameters are estimations of the leaked model's weights (as detailed in Algorithm 1 in Appendix C), whereas the parameters of CoarseGIT represent a black-box approximation.

Figure 3 illustrates the L2 distance curve between the attack model's weights and the leaked model's weights, alongside the MSE between the reconstructed inputs and the ground truth inputs. As shown in the figure, when FineGIT converges, its weights align closely with the ground truth weights. This convergence highlights the effectiveness of FineGIT in extracting weight information from the leaked model.

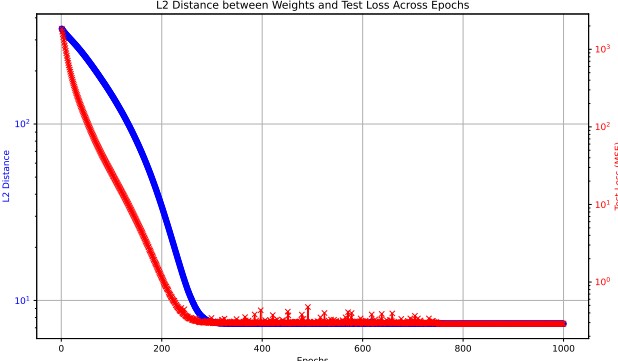

Figure 3: The red curve represents L2 distance between weights of the attack model and the leaked model. The blue curve represents MSE between reconstructed input and the ground truth input. The experiment is conducted on leaked model with two convolutional layers for 1000 epochs. The dataset is CIFAR10, 5000 samples are leaked to the attacker. These curves show the trend of L2 distance during training.

## 6 CONCLUSIONS

This work introduces the *Generative Gradient Inversion Transcript (GIT)*, a method for reconstructing training data in distributed learning by exploiting gradient leakage. We formulate and solve a reconstruction system that leverages gradients to recursively reconstruct the hidden layer neuron outputs, based on the back propagation. Our framework is generic and considers different categories of layers and network topologies. Our experiments demonstrate the effectiveness of our proposed methods: compared with using a fixed architecture as the generative model for reconstruction, GIT is more adaptive to different architectures of the leaked models. GIT has competitive performance in various scenarios, including noisy gradients and limited amount of training data. Our future work will focus on mitigating the overfitting issue to further improve the performance of generative reconstruction methods.

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

## A  NOTATION

| | |
|---|---|
| $\mathcal{L}$ | Loss objective function |
| $\sigma$ | An activation function |
| $\boldsymbol{z}_i$ | pre-activation output of $i$-th hidden layer in a neural network |
| $\boldsymbol{a}_i$ | post-activation output of of $i$-th hidden layer in a neural network |
| $\mathbf{W}_i$ | A weight tensor of $i$-th layer in a neural network |
| $\boldsymbol{g}_i$ | A gradient tensor of $i$-th layer in a neural network |
| $B$ | Batch size |
| $N$ | Number of hidden layers in a neural network |
| $\boldsymbol{x}$ | A single data batch |
| $\mathbf{X}$ | A series of data batches |
| $y$ | Label of a data sample |
| $\boldsymbol{y}$ | Labels of a single data batch |
| $\mathbf{Y}$ | Labels of a series of data batches |
| $(.)^{(i)}$ | The $i$-th sample in the set |
| $x^+$ | Moore-Penrose inverse of each of $x$'s subspace via the first dimension |
| $\otimes$ | Tensor Multiplification |
| $\odot$ | Broadcast row-wise product |

## B  EXPERIMENT CONFIGURATION

In our experiments described in Section 5, we reconstruct training data using a five-layer LeNet and a twenty-layer ResNet, both employing a kernel size of 5 and with each output channel set to 12. The last layers of both models are fully connected layers. In ResNet, every two convolutional layers form a basic block, connected by skip connections.

For the generative approach using a Multi-Layer Perceptron (MLP), we design the hidden size to be 3000, with a total of three hidden layers, consistent with the architecture proposed by Wu et al. (2023). In this experiment, we utilize the CoarseGIT model instead of FineGIT. The reconstruction results are presented on the test set of CIFAR-10, showcasing the first eight images to illustrate the visible reconstruction performance.

## C Algorithm Pseudocode

---

**Algorithm 1** Generative Gradient Inverse Transcript (GIT)

---

1: **Setup:** Set network width for layer-$i$ in leaked model as $\{d_i\}_{i=1}^{N-1}$. With $M$ known batches of training data-gradient pairs for distributed learning $D = \{(\mathbf{X}^1, \boldsymbol{y}^1), (\mathbf{X}^2, \boldsymbol{y}^2), \ldots, (\mathbf{X}^M, \boldsymbol{y}^M)\}$, we have shared gradients $\boldsymbol{g}_i^m = \nabla_{\mathbf{W}_i}\mathcal{L}(\mathbf{X}^m; \boldsymbol{y}^m)$, for $i = 1, \ldots, N$ and $m = 1, \ldots, M$; as well as update of final layer's bias $\{\frac{\partial\mathcal{L}}{\partial\boldsymbol{b}_N}\}^m$ for $m$-th batch.

2: **Initialization:** Current GIT model parameters $\Theta := \{\mathbf{W}_1, \mathbf{W}_2, \ldots, \mathbf{W}_N\}$ are initialized randomly as:
$$\mathbf{W}_i \sim \mathcal{N}(0, \sigma^2), \quad i = 1, 2, \ldots, N$$

3: **Training:**

4: Set $\epsilon$ as the learning rate. GIT is trained on $D$ for $E$ epochs.

5: **for** each epoch $e = 1$ to $E$ **do**

6:     **for** each batch $m = 1$ to $M$ **do**

7:         Input: Gradients $\boldsymbol{g}_i^m$, for $i = 1, \ldots, N-1$.

8:         Compute the embedding $\boldsymbol{a}_{N-1}^T = g_N^m \left(\{\frac{\partial\mathcal{L}}{\partial\boldsymbol{b}_N}\}^m\right)^{-1}$.

9:         **for** each layer $i = N-1$ to $1$ **do**

10:             $\boldsymbol{a}_i'(j) = \begin{cases} 1, & \text{if } \boldsymbol{a}_i(j) > 0 \\ 0, & \text{if } \boldsymbol{a}_i(j) = 0 \end{cases} \quad \forall j \in \{1, 2, ..., d_i\}$

11:             $\boldsymbol{a}_{i-1}^T = \boldsymbol{a}_i^T \otimes (\boldsymbol{g}_{i+1}^m)^{-1} \otimes \left(\mathbf{W}_{i+1}^T \odot \boldsymbol{a}_i'\right)^{-1} \otimes \boldsymbol{g}_i^m,$

12:         **end for**

13:         Output: Recovered estimated input $\hat{\mathbf{X}}^m = \boldsymbol{a}_0$.

14:         Compute $\mathcal{L}_{GIT} = ||\hat{\mathbf{X}}^m - \mathbf{X}^m||^2$ as the reconstruction error

15:         Update model parameters $\mathbf{W}_i$: $\mathbf{W}_i \leftarrow \mathbf{W}_i - \epsilon\nabla_{\mathbf{W}_i}\mathcal{L}_{GIT}(\boldsymbol{g}^m; \mathbf{X}^m)$, $i = 1, 2, \ldots, N$

16:     **end for**

17: **end for**

18: **Reconstruction:** To reconstruct a batch of unknown training data $\mathbf{X}$ for distributed learning with corresponding gradient $\boldsymbol{g}_i$, for $i = 1, \ldots, N$.

19: Input: Gradients $\boldsymbol{g}_i$, for $i = 1, \ldots, N-1$

20: Compute the embedding $\boldsymbol{a}_{N-1}^T = g_N \left(\frac{\partial\mathcal{L}}{\partial\boldsymbol{b}_N}\right)^{-1}$.

21: **for** each layer $i = N-1$ to $1$ **do**

22:     $\boldsymbol{a}_i'(j) = \begin{cases} 1, & \text{if } \boldsymbol{a}_i(j) > 0 \\ 0, & \text{if } \boldsymbol{a}_i(j) = 0 \end{cases} \quad \forall j \in \{1, 2, ..., d_i\}$

23:     $\boldsymbol{a}_{i-1}^T = \boldsymbol{a}_i^T \otimes (\boldsymbol{g}_{i+1})^{-1} \otimes \left(\mathbf{W}_{i+1}^T \odot \boldsymbol{a}_i'\right)^{-1} \otimes \boldsymbol{g}_i,$

24: **end for**

25: Output: Recovered estimated input $\hat{\mathbf{X}} = \boldsymbol{a}_0$.

---

# D GIT ARCHITECTURE FOR LEAKED MODEL WITH SKIP CONNECTIONS

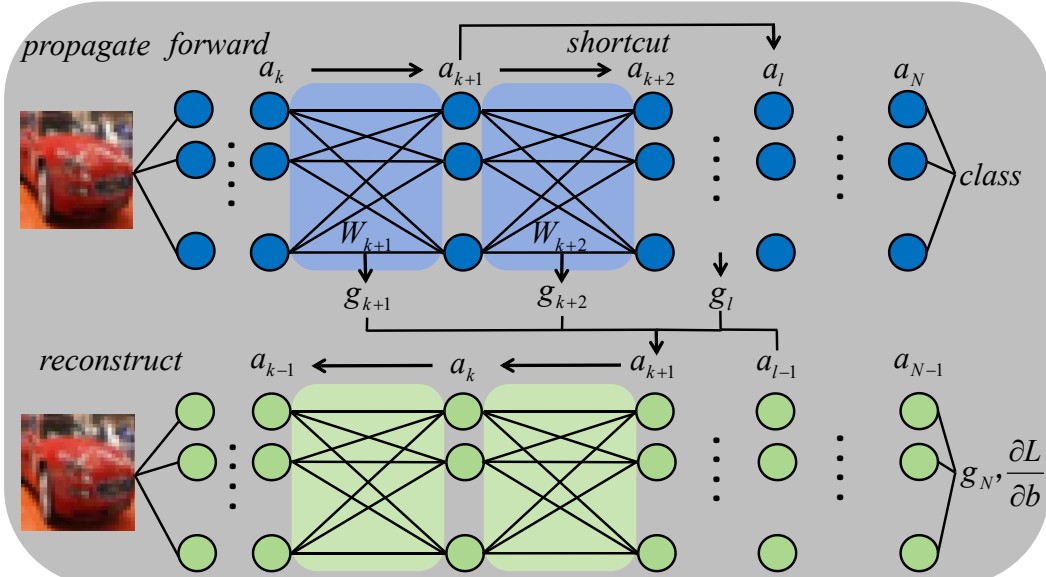

Figure 4: **(Top half)** The leaked model which leaks the gradient to the attakers. **(Bottom half)** The threat model constructed by Inverse Gradient Transcript (GIT) based on the approximation (6). The threat model is a generative model utilizing the leaked gradients to reconstruct the training mini-batch data. In FineGIT mode, we estimate $a_k$ based on the approximation (6) with unknown variables as trainable parameters. In CoarseGIT mode, we use an MLP to estimate $a_k$ with the gradient and activation estimation based on (6) as the input.

# E    ADDITIONAL EXPERIMENTAL RESULTS

## E.1    RECONSTRUCTED IMAGES FOR DIFFERENT METHODS ON CIFAR10

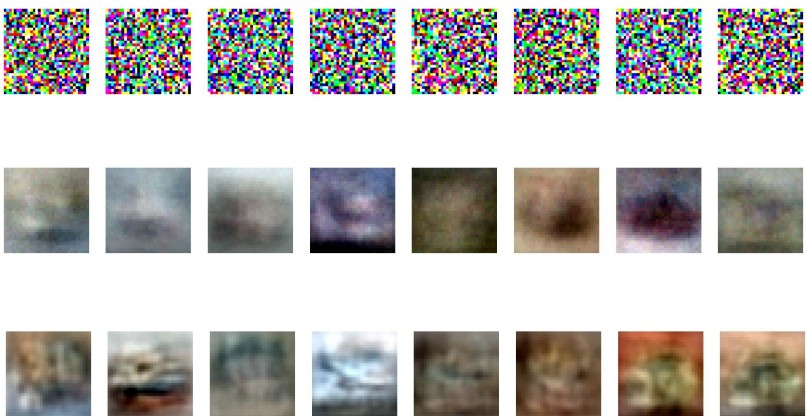

Figure 5: Comparison the first 8 reconstructed images in CIFAR-10 test set when using different reconstruction method. The leaked model is ResNet and batch size is 1. **(From top to bottom)** DLG, generative approach utilizing MLP, generative approach utilizing GIT. The results show that both DLG and the generative approach using MLP fail to recover reasonable images on ResNet, while GIT is able to reconstruct some features of the ground truth images.

## E.2    EXPERIMENTAL RESULTS FOR GIT ON TINYIMAGENET-200

Table 6: MSE for reconstructed TinyImageNet with different batch sizes and model types.

| Leaked Model | Metrics | Batch Size = 1 | Batch Size = 2 | Batch Size = 4 |
|---|---|---|---|---|
| LeNet 5 | **MSE** | 0.0317 | 0.0437 | 0.0509 |
| | **PSNR** | 14.99 | 13.60 | 12.93 |
| ResNet 20 | **MSE** | 0.0983 | 0.1147 | 0.1274 |
| | **PSNR** | 10.07 | 9.40 | 8.95 |

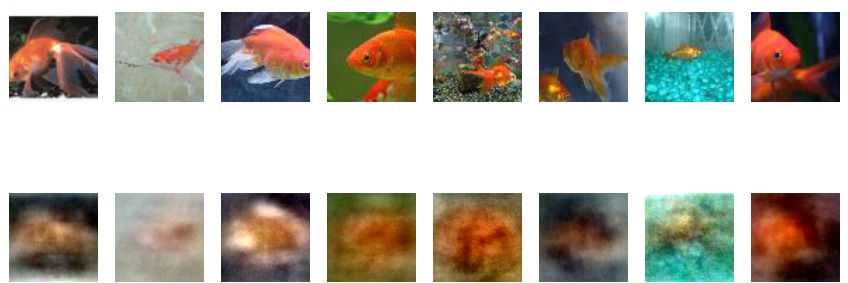

Figure 6: The best MSE of the reconstructed images are $0.0317 \pm 0.0003$. And the corresponding first 8 reconstructed images (bottom) and ground truth images (top) of TinyImageNet-200, with 10000 training samples.

