# OpenReview forum: "Gradient Inversion Transcript: A Generative Model to Reconstruct Training Data by Gradient Leakage"
_ICLR.cc/2025/Conference — Submitted to ICLR 2025_

### Official Review · Reviewer_Bm4v · 2024-10-29

**Soundness:** 3
**Presentation:** 3
**Contribution:** 3
**Rating:** 6
**Confidence:** 4

**Summary:**

The authors a method called propose Gradient Inversion Transcript (GIT) relying on generative models to reconstruct the input. GIT does not rely on model weights but only needs the model architecture. The authors claim that this makes it more applicable to the real world setting. Further, their method adaptively chooses an architecture for the generative method.

Their experiments where conducted over LeNet and ResNet for batch sizes of up to 4 over the CIFAR-10 dataset, demonstrating the effectiveness against baselines. Further, they conduct an ablation study over the number of training samples used to train their model.

**Strengths:**

- Gradient inversion attacks are a crucial way to investigate the privacy of federated learning methods.
- The training approach as formalized in Algorithm 1 seems interesting and novel.
- It is good to demonstrate the effectiveness of this method in the setting of noisy gradients.
- The paper was overall easy to read.

**Weaknesses:**

- The threat model appears to be not well motivated: It is unclear in which scenario an attacker has access to the gradient updates but not the  model weights. In other words - what is the incentive to not prevent sending gradients to 3rd parties that do not contribute to training? Following that - how is it more practical for the attacker to have access to input-gradient pairs? Where would they come from in practice.
- It is unclear if the claim that some assumptions are stronger holds here in practice? Specifically, what is a stronger assumption - having sufficient training data or the network weights? There are some approaches that do not rely on priors on the dataset, don't need multiple gradient querying, no labels and are exact (see [1] and [2]). Also the math appears related.

Further:
- L33 - the sentence after "federated learning (FL)" does not seem complete.
- L48 - the use of the term "threat model" for the model doing the attack is unfortunate, unnecessarily overloading there this term. This is problematic because the authors claim relevance of a weaker threat model where the attacker does not have access to model weights.
- Table 3 - maybe the best reported number could be bolted.

Citations:
- 1) Dimitrov et al. "SPEAR: Exact Gradient Inversion of Batches in Federated Learning", https://arxiv.org/abs/2403.03945
- 2) Petrov et al. "DAGER: Exact Gradient Inversion for Large Language Models", https://arxiv.org/abs/2405.15586

**Questions:**

- How does it generalize to Federated learning settings like federated averaging?
- What Architectures? Appears that only linear and skip connections can be dealt with (Section 3). What about say transformer architectures?
- In which circumstances is the threat model realistic? Access to lost of training data but not the model weights?
- Given the number of training samples of inputs and gradients - could one adapt gradient matching techniques to weight matching techniques to reconstruct weights?
- How does your approach scale to larger batch sizes and more complex datasets? How do you scale to deep networks? Does the reviewer suppose correctly that this is more difficult?

Typos
- 194: "Since both $\sigma$ and $\sigma$" appears wrong
- 245 - there should be $\{\}$ around $g_i$.

---

> ### Author Response · Authors · 2024-11-23
>
> **Answer to Weakness 1. Practical Scenario**
>
> To the best of our knowledge, most existing works of training data reconstruction also assume that the attacker is able to intercept gradients transmitted from client nodes to the server but not the model weights.  [2], [3], [4] and [5] assume gradient leakage and model querying, which can be seen as a gray box. [1] assumes gradient leakage and part of training data, which can be seen as a black box. **These two assumptions are both reasonable and based on different attacker's access to the target model, gray box with multiple querying and black box with several input-gradient pairs, are both able to steal information from the model.**
>
> Weights leakage, in contrast, represents a stronger assumption as it reveals the complete model information. This essentially transforms the Federated Learning (FL) model into a white-box scenario, enabling the derivation of a closed-form reconstruction algorithm ([6], for example, assumes both gradient leakage and weights leakage, regarded as a white box, where the model information is totally leaked).
>
> In our work we consider a more relaxed assumption compared to weights leakage, where attacker intercepts gradients transmitted from client nodes to the central server and additionally breaches one client node to gain access to its local dataset. **In addition, it is noteworthy that if model querying is allowed, we can use public data to generate input-gradient pairs and there is no need for obtaining the real leaked data.**
>
> **Answer to Weakness 2. Stronger Assumption**
>
> It is challenging to determine which assumption is stronger, as the strength of an assumption depends on the amount of leaked training data. With sufficient training data, a closer estimation of the model's information can be achieved, albeit at a higher cost.
>
> DAGER primarily focuses on large language models (LLMs), which is interesting but falls outside the scope of this paper. On the other hand, SPEAR operates under stronger assumptions, requiring the gradients of biases for all layers and no activations are allowed except in the final layer. However, it addresses a critical issue commonly faced in training data reconstruction from gradient leakage: the information loss caused by large batch sizes. While SPEAR is an instructive study, large batch size reconstruction is not the focus of our work, and the assumption of attacker's access is too strong in SPEAR compared with our settings.
>
> Instead, we concentrate on achieving optimal recovery performance with the same model complexity under weaker assumptions by designing a robust and efficient attack model structure.
>
> **Answer to Editorial Comments**
>
> We agree that changing "threat model" to "attack model" or directly "GIT" would better reflects its meaning in the given context. Other modifications for editorial comments please refer to revision.
>
> **Answer to Question 1. Scenario**
>
> In federated learning settings like Federated Averaging (FedAvg), during a communication round, each client trains the model on its local dataset. The transmitted gradients are then intercepted by the attacker, and one client node is breached to the attacker so that part of the training data is leaked.
>
> **Answer to Question 2. Architecture**
>
> Based on the backpropagation, our methods (both CoarseGIT and FineGIT) can be applied to general neural architectures as long as backpropagation can be applied. In addition, we need to point out that the architecture of the generative model is not a direct inverse of the target model but based on the derivations like Equation (4) and (5). To the best of our knowledge, transformer has not been used in previous works as the target model for gradient inversion under our settings. Therefore, we do not include it for the lack of baselines.
>
> **Answer to Question 3. Assumptions Comparison**
>
> Refer to **Answer to Weakness 2. Stronger Assumption.**
>
> **Answer to Question 4. Weights Reconstruction**
>
> It's a constructive research direction, we also consider using gradient matching to reconstruct weights, which is the inverse of reconstructing inputs by gradient matching.
>
> But model querying is necessary to implement gradient matching approach while we assume the attacker has no access to backpropagation queries in this work.

---

> > ### Author Response · Authors · 2024-11-23
> >
> > **Answer to Question 5. Complementary Experiments**
> >
> > The complementary experiments for additional dataset TinyImageNet-200 is shown in the table below. The reconstructed images are shown in  Appendix E.2 in revision, with its best MSE of 0.0317 $\pm$ 0.0003. The results demonstrate that high-frequency information, including object contours and background details, is effectively recovered. Although the MSE and visual quality are lower compared to CIFAR10 under the same configuration, reconstructing data from higher-resolution images poses a significant challenge. Notably, resolutions larger than CIFAR10 have not been explored in baseline methods MLP and DLG.
> >
> > | **Leaked Model**    | **Metrics** | **Batch Size = 1** | **Batch Size = 2** | **Batch Size = 4** |
> > |----------------------|-------------|--------------------|--------------------|--------------------|
> > | **LeNet 5**         | **MSE**     | 0.0317             | 0.0437             | 0.0509             |
> > |                      | **PSNR**    | 14.99              | 13.60              | 12.93              |
> > |----------------------|-------------|--------------------|--------------------|--------------------|
> > | **ResNet 20**       | **MSE**     | 0.0983             | 0.1147             | 0.1274             |
> > |                      | **PSNR**    | 10.07              | 9.40               | 8.95               |
> >
> > **References**
> >
> > [1] Ruihan Wu, Xiangyu Chen, Chuan Guo, and Kilian Q Weinberger. Learning to invert: Simple adaptive attacks for gradient inversion in federated learning. In Uncertainty in Artificial Intelligence, pp. 2293–2303. PMLR, 2023.
> >
> > [2] Ligeng Zhu, Zhijian Liu, and Song Han. Deep leakage from gradients. In H. Wallach, H. Larochelle, A. Beygelzimer, F. d'Alch´e-Buc, E. Fox, and R. Garnett (eds.), Advances in Neural Information Processing Systems, volume 32. Curran Associates, Inc., 2019.
> >
> > [3] Bo Zhao, Konda Reddy Mopuri, and Hakan Bilen. idlg: Improved deep leakage from gradients. arXiv preprint arXiv:2001.02610, 2020.
> >
> > [4] Jonas Geiping, Hartmut Bauermeister, Hannah Dr¨oge, and Michael Moeller. Inverting gradients-how easy is it to break privacy in federated learning? Advances in neural information processing systems, 33:16937–16947, 2020.
> >
> > [5] Hongxu Yin, Arun Mallya, Arash Vahdat, Jose M Alvarez, Jan Kautz, and Pavlo Molchanov. See through gradients: Image batch recovery via gradinversion. In Proceedings of the IEEE/CVF conference on computer vision and pattern recognition, pp. 16337–16346, 2021.
> >
> > [6] Junyi Zhu and Matthew Blaschko. R-gap: Recursive gradient attack on privacy. arXiv preprint
> > arXiv:2010.07733, 2020.

---

> > > ### Author Response · Authors · 2024-11-29
> > >
> > > This is a gentle reminder that the discussion period is nearing its end. We kindly ask you to review our response and let us know if it sufficiently addresses your concerns. Your feedback is highly valuable to us. For your convenience, we have summarized some key points from our discussions with other reviewers below:
> > >
> > > 1. In response to reviewer TtUQ, we conducted complementary experiments on the distance between weights using FineGIT, **which highlights the effectiveness of GIT in extracting weight information from the leaked model**. More details can be found in Section 5.5 of the revised manuscript.
> > >
> > > 2. In line with reviewer tY9t’s suggestion, we performed additional experiments on generative approaches, incorporating UNet as an additional baseline and PSNR as an additional metric. Table 5 in the revision presents a quantitative comparison of GIT with other generative methods, using both fixed MLP and UNet architectures, which share a similar number of parameters to GIT.
> > >
> > > 3. To highlight the advantages of CoarseGIT over a simple MLP, **we conducted ablation studies by varying the depth and width of the MLP. These experiments demonstrate that our proposed GIT model outperforms all configurations of the baseline generative methods, showcasing its superior performance under fair conditions, where computational complexity is kept equal across models**. More details are illustrated in section 5.4 of the revised manuscript.

---

### Official Review · Reviewer_qycS · 2024-10-31

**Soundness:** 2
**Presentation:** 1
**Contribution:** 2
**Rating:** 3
**Confidence:** 5

**Summary:**

This paper proposes GIT (Gradient Inversion Transcript) to reconstruct training data from gradient information in distributed learning settings. Unlike previous methods that require model parameters or repeated gradient queries, GIT only needs model architecture information and works offline. It adaptively designs the generative network's architecture based on the target model's structure, theoretically derived from backpropagation equations. The authors demonstrate better reconstruction accuracy compared to baselines, especially for deeper models.

**Strengths:**

1. The adaptive architecture of the generative model, mirroring the target model's structure, allows for more effective exploitation of gradient information compared to fixed-architecture generative methods.
2. The paper considers a practical attack scenario where the adversary only has access to shared gradients without knowledge of model parameters, labels, or the ability to query the model. This aligns with real-world constraints faced by attackers.
3. The method has some theoretical analysis of backpropagation, providing a stronger justification for the design choices compared to purely heuristic approaches.

**Weaknesses:**

1. Overfitting issues: The authors acknowledge significant overfitting problems but don't provide solutions
2. Experimental scope: Experiments are primarily limited to CIFAR-10 and two specific network architectures (LeNet and ResNet).
3. Baseline comparison: The paper lacks a comprehensive comparison with state-of-the-art methods, making it difficult to conclude the superiority of GIT over existing approaches.

**Questions:**

1. Theoretical Foundation of the approximation: The paper uses the pseudo-inverse for approximation. While the pseudo-inverse offers a solution for non-invertible matrices, the paper lacks a theoretical justification for its application in this specific context. Furthermore, are there any analytical bounds on the error introduced by this approximation, and how does this error may propagate through the reconstruction process?
2. Practical Applicability and Advantages of MLPs: Given the acknowledged numerical instability of directly computing approximation, the paper often resorts to using MLPs for approximation. This raises questions about the practical advantages of GIT over simply training a larger, more complex MLP directly from gradients to input data. If MLPs are primarily used, what specific advantages does GIT retain over other generative methods? Any explanation on why GIT may still have an edge over MLP methods?

---

> ### Author Response · Authors · 2024-11-23
>
> **Answer to Weakness 1. Overfitting**
>
> Overfitting is not the major issue contributing to the performance difference, because both GIT and MLP have such issue. There are some general solutions to mitigate overfitting, such as data augmentation, but it is out of the scope of this paper.
>
> **Answer to Weakness 2. Experimental Scope**
>
> The complementary experiments for additional dataset TinyImageNet-200 is shown in the table below. The reconstructed images are shown in  Appendix E.2 in revision, with its best MSE of 0.0317 $\pm$ 0.0003. The results demonstrate that high-frequency information, including object contours and background details, is effectively recovered. Although the MSE and visual quality are lower compared to CIFAR10 under the same configuration, reconstructing data from higher-resolution images poses a significant challenge. Notably, resolutions larger than CIFAR10 have not been explored in baseline methods MLP and DLG.
>
> | **Leaked Model**    | **Metrics** | **Batch Size = 1** | **Batch Size = 2** | **Batch Size = 4** |
> |----------------------|-------------|--------------------|--------------------|--------------------|
> | **LeNet 5**         | **MSE**     | 0.0317             | 0.0437             | 0.0509             |
> |                      | **PSNR**    | 14.99              | 13.60              | 12.93              |
> |----------------------|-------------|--------------------|--------------------|--------------------|
> | **ResNet 20**       | **MSE**     | 0.0983             | 0.1147             | 0.1274             |
> |                      | **PSNR**    | 10.07              | 9.40               | 8.95               |
>
> **Answer to Weakness 3. Baseline comparison**
>
> Gradient matching approaches have different assumptions compared to ours, so we conduct experiments only on the most popular framework DLG as a reference.
>
> For a more comprehensive comparison and reference of existing training data reconstruction approaches, we conduct complementary experiments on generative approach with UNet as the additional baselines. The table below (also in Table to in the revision) shows quantitative comparison of GIT with prior works on different networks and batch sizes. We use MLP \& UNet to represent the generative method using a fixed MLP \& UNet architecture, which shares similar number of parameters to our proposed method. The numbers in the table represent the MSE \& PSNR between the reconstructed data and the ground truth on the test set. We use $10000$ samples to train generative models.
>
> | **Leaked Model**        | **Method** | **Metrics** | **Batch Size = 1** | **Batch Size = 2** | **Batch Size = 4** |
> |--------------------------|------------|-------------|--------------------|--------------------|--------------------|
> | **LeNet (5 layers)**     | **DLG**    | **MSE**     | **0.0008**         | 0.0472             | 0.0975             |
> |                          |            | **PSNR**    | **30.97**          | 13.26              | 10.11              |
> |                          | **MLP**    | **MSE**     | 0.0241             | 0.0332             | 0.0571             |
> |                          |            | **PSNR**    | 16.18              | 14.79              | 12.43              |
> |                          | **GIT**    | **MSE**     | 0.0099             | **0.0122**         | **0.0254**         |
> |                          |            | **PSNR**    | 20.04              | **19.14**          | **15.95**          |
> |                          | **UNet**   | **MSE**     | 0.0316             | 0.0393             | 0.0435             |
> |                          |            | **PSNR**    | 15.00              | 14.06              | 13.62              |
> | **ResNet (20 layers)**   | **DLG**    | **MSE**     | 0.1202             | 0.1347             | 0.1365             |
> |                          |            | **PSNR**    | 9.20               | 8.71               | 8.65               |
> |                          | **MLP**    | **MSE**     | 0.0354             | 0.0473             | 0.0589             |
> |                          |            | **PSNR**    | 14.51              | 13.25              | 12.30              |
> |                          | **GIT**    | **MSE**     | **0.0193**         | **0.0246**         | **0.0388**         |
> |                          |            | **PSNR**    | **17.14**          | **16.09**          | **14.11**          |
> |                          | **UNet**   | **MSE**     | 0.0515             | 0.0560             | 0.0619             |
> |                          |            | **PSNR**    | 12.88              | 12.52              | 12.02              |

---

> > ### Author Response · Authors · 2024-11-23
> >
> > **Answer to Question 1**
> >
> > 1. Error Analysis in Pseudo-inverse Calculation
> >
> > Given a matrix $A \in \mathbb{R}^{m \times n}$, the true pseudo-inverse $A^{+}$ is based on the limit defination:
> > \begin{equation}
> > \begin{aligned}
> > A^+ = \lim_{\lambda \to 0^+} (A^T A + \lambda I)^{-1} A^T,
> > \end{aligned}
> > \end{equation}
> > where $\lambda$ is a small regularization parameter.
> >
> > But in implementation, pseudo-inverse is calculated using Singular Value Decomposition (SVD):
> > \begin{equation}
> > \begin{aligned}
> > A = U \Sigma V^T, \quad A^+_{\text{SVD}} = V \Sigma^+ U^T,
> > \end{aligned}
> > \end{equation}
> > where $\Sigma^+$ is obtained by inverting the nonzero singular values in $\Sigma$.
> >
> > The error between the pseudo-inverse computed using SVD and the true pseudo-inverse can be expressed as:
> > \begin{equation}
> > \begin{aligned}
> > ||A^+_{\text{SVD}} - A^+|| \leq \epsilon_t + \epsilon_r,
> > \end{aligned}
> > \end{equation}
> >
> > where $\epsilon_t$ is the truncation error caused by ignoring small singular values in SVD to avoid instability, and $\epsilon_r$ is the regularization error introduced by using a nonzero regularization parameter $\lambda$. Both errors depend on the condition number of $A$.
> >
> > 2. The Propagation of Error in the Attack Model.
> >
> > In the recursive equation of GIT (See equation (4)), the error $\epsilon_{a_i}$ of the reconstructed output $a_i$ in the previous layer will be propagated to the reconstructed output of next layer $a_{i-1}$.
> > \begin{equation}
> > \begin{aligned}
> > ||\epsilon_{a_{i-1}}|| \leq ||\epsilon_{a_i}|| \cdot \sigma_{max}( g_{i+1}^+ \otimes (W_{i + 1}^T \odot \sigma_i'(z_i))^+ \otimes g_i) \\
> > \leq ||\epsilon_{a_i}|| \cdot \sigma_{max}(g_{i+1}^+ )\cdot \sigma_{max}((W_{i + 1}^T \odot \sigma_i'(z_i))^+) \cdot  \sigma_{max}(g_i)
> > \end{aligned}
> > \end{equation}
> >
> > In the experiments setting, since gradient matrices are typically sparse and consist of small values, the maximum eigenvalues of above metrics are generally small, effectively preventing explosion during propagation.
> >
> > **Answer to Question 2**
> >
> > Our method is distinct from a standard MLP. Even in the framework of CoarseGIT, we roughly keep the structure of GIT which depends on the architecture of the target model except that we use MLP modules to approximate the psuedo-inverse. This is significantly different from simply employing a MLP as the generative model, even in the case when the target model is small.
> >
> > To further demonstrate the advantage of our model over a simple MLP, we use an MLP with the same parameters as a baseline for comparison. We conduct ablation studies by varying the depth and width of an MLP. The results are shown in the table below (also in Table 5 in rivision). All ablation studies are performed with a LeNet FL model and a batch size of 1.
> >
> > In the generative approach with a fixed MLP, a three-layer structure is adopted, each with 1000 hidden units. The first ablation study keeps the model depth constant but increases the depth of each layer to match the number of parameters of GIT, allowing us to evaluate the differences between our method and a standard MLP under the same model complexity. The second experiment maintains the hidden size of each MLP layer but increases the model depth to match that of GIT. However, in these configurations, all gradients are flattened and input from the first layer, rather than fed incrementally through layers as in Coarse GIT. **The results show that our proposed GIT model performs better than all configurations of baselines, highlighting its superior performance under fair conditions of equal computational complexity.**
> >
> > Furthermore, GIT is not merely a reversed version of the leaked model. Instead, it incorporates skip connections and activation inversion structures, specifically designed and adapted based on insights derived from the leaked model.
> >
> > | **MLP**                | **MLP with Fixed Depth** | **MLP withFixed Width** |    **GIT**         |
> > |------------------------|---------------------------------|--------------------------------|-------------------------|
> > | 0.0241 ± 0.0003  | 0.0224 ± 0.0003            | 0.0129 ± 0.0004            | **0.0099 ± 0.0001** |

---

> > > ### Author Response · Authors · 2024-11-29
> > >
> > > This is a gentle reminder that the discussion period is nearing its end. We kindly ask you to review our response and let us know if it sufficiently addresses your concerns. Your feedback is highly valuable to us. For your convenience, we have summarized some key points from our discussions with other reviewers below:
> > >
> > > 1. In response to reviewer TtUQ, we conducted complementary experiments on the distance between weights using FineGIT, which highlights the effectiveness of GIT in extracting weight information from the leaked model. More details can be found in Section 5.5 of the revised manuscript.
> > >
> > > 2. In line with reviewer tY9t’s suggestion, we performed additional experiments on generative approaches, incorporating UNet as an additional baseline and PSNR as an additional metric. Table 5 in the revision presents a quantitative comparison of GIT with other generative methods, using both fixed MLP and UNet architectures, which share a similar number of parameters to GIT.

---

### Official Review · Reviewer_tY9t · 2024-11-01

**Soundness:** 3
**Presentation:** 3
**Contribution:** 2
**Rating:** 5
**Confidence:** 3

**Summary:**

The paper proposes the Gradient Inversion Transcript (GIT), focusing on its theoretical basis and empirical application in reconstructing training data with generative model. The setting is that the model could only get access to the leaked gradient. The central concept relies on a mathematical extension based on Equation 4 and gradient-based back-propagation. The paper conducts experiments on the CIFAR-10 dataset and reconstructs images with lower MSE error than the baselines.

**Strengths:**

GIT introduces a theoretically informed generative model tailored to the target model’s architecture, making it expirically better compared to traditional fixed architectures.

**Weaknesses:**

Several critical weaknesses are listed below:

[Theoretical Side]:

The theoretical benefits from the paper is limited. GIT is essentially based on Equation 4, and Equation 4 is a straightforward extension and observation from the gradient back-propagation.

[Empirical side]:

1.The empirical experiment is limited. Experiments are only conducted on cifar10 dataset.

2.Lack of important implementation details in the experiments.
What is the size of the fixed MLP layers? Does it have the same number of parameters of the NN discovered by GIT for fairness?
Also the implementation of the baselines are blank. There is no images generated by the baseline shown in the paper.

3.Lack of important experiment results in the experiments. What is the optimized neural network architecture? How similar is it towards the target leaked model? How to define a metric to illustrate the effectiveness of the optimized neural network structure?

**Questions:**

1.The paper does not consider the UNet structure as a baseline, Why UNet leverages priors from public data? It actually should be a wel-established widely used baseline.

---

> ### Author Response · Authors · 2024-11-23
>
> **Answer to Weakness. Theoretical Side**
>
> Although Equation 4 is based on the back propagation, it is generic and applicable to general model architectures. \textbf{To the best of our knowledge, this is the first generative model to conduct gradient inversion where the architecture of the generator adapts to the leaked model.} This improves not only the performance but also the interpretability of the method. Moreover, the inverse of activation function and the derived computational graph is not a direct inverse of the original model. For example, Equation (4) use $g_{i}$, $g_{i + 1}$ and $a_i$ to reconstruct $a_{i - 1}$, which is different computational graph from back propagation. Equation (5) is a similar example as well.
>
> **Answer to Weakness. Empirical Side 1. Different datasets**
>
> The complementary experiments for additional dataset TinyImageNet-200 is shown in the table below. The reconstructed images are shown in  Appendix E.2 in revision, with its best MSE of 0.0317 $\pm$ 0.0003. The results demonstrate that high-frequency information, including object contours and background details, is effectively recovered. Although the MSE and visual quality are lower compared to CIFAR10 under the same configuration, reconstructing data from higher-resolution images poses a significant challenge. Notably, resolutions larger than CIFAR10 have not been explored in baseline methods MLP and DLG.
>
> | **Leaked Model**    | **Metrics** | **Batch Size = 1** | **Batch Size = 2** | **Batch Size = 4** |
> |----------------------|-------------|--------------------|--------------------|--------------------|
> | **LeNet 5**         | **MSE**     | 0.0317             | 0.0437             | 0.0509             |
> |                      | **PSNR**    | 14.99              | 13.60              | 12.93              |
> |----------------------|-------------|--------------------|--------------------|--------------------|
> | **ResNet 20**       | **MSE**     | 0.0983             | 0.1147             | 0.1274             |
> |                      | **PSNR**    | 10.07              | 9.40               | 8.95               |
>
> **Answer to Weakness. Empirical Side 2. Implementation Details**
>
> For detailed implementation, please refer to Appendix B - Experiment Configuration in the paper. The images reconstructed by baselines and our proposed method are both shown in Figure 5 \& 6 in Appendix E in revision.
>
> The ablation studies of number of parameters are shown in the table below (also in Table 5 in the paper).
>
> Since GIT learns to invert gradients based on the architecture dependent on the model under attack, its model complexity varies for different FL models and differs from the generative approach that employs a fixed architecture. Generally speaking, during inference, the complexity of running GIT is proportional to running the model under attack, because their architectures are related.
>
> To understand this dependence, we conduct ablation studies by varying the depth and width of an MLP. The results are shown in the table below. All ablation studies are performed with a LeNet FL model and a batch size of 1.
> In the generative approach with a fixed MLP, a three-layer structure is adopted, each with 1000 hidden units. The first ablation study keeps the model depth constant but increases the depth of each layer to match the number of parameters of GIT, allowing us to evaluate the differences between our method and a standard MLP under the same model complexity. The second experiment maintains the hidden size of each MLP layer but increases the model depth to match that of GIT. However, in these configurations, all gradients are flattened and input from the first layer, rather than fed incrementally through layers as in Coarse GIT. **The results show that our proposed GIT model
> performs better than all configurations of baselines, highlighting its superior performance under fair conditions of equal computational complexity.**
>
> | **MLP**                | **MLP with Fixed Depth** | **MLP withFixed Width** |    **GIT**         |
> |------------------------|---------------------------------|--------------------------------|-------------------------|
> | 0.0241 ± 0.0003  | 0.0224 ± 0.0003            | 0.0129 ± 0.0004            | **0.0099 ± 0.0001** |

---

> > ### Author Response · Authors · 2024-11-23
> >
> > **Answer to Weakness. Empirical Side 3. Experimental Results**
> >
> > The architecture of leaked model and the optimized attack model are shown in Figure 1 (for Coarese GIT) and Algorithm 1 (for Fine GIT).
> >
> > We use the MSE between the reconstructed images and the ground truth images as the metrics to measure the effectiveness of our proposed reconstruction framework in the tables in our manuscript. In addition, complementary experiment measures L2 distance between the weights of optimized neural network and leaked neural network, which are also conducted as the metric.
> >
> > The experiment is conducted on FineGIT, as its parameters are estimations of the leaked model's weights (as detailed in Algorithm 1 in Appendix C), whereas the parameters of CoarseGIT represent a black-box approximation.
> >
> > **Figure 2 in the paper illustrates the L2 distance curve between the attack model's weights and the leaked model's weights, alongside the MSE between the reconstructed inputs and the ground truth inputs.** As shown in the figure, when FineGIT converges, its weights align closely with the ground truth weights. This convergence highlights the effectiveness of FineGIT in extracting weight information from the leaked model.
> >
> > **Answer to Questions**
> >
> > UNet structure is not considered as a baseline in this paper since there is no existing work adopting UNet as a generative model to reconstruct training data. In one previous work (see [1]), UNet is used to generate image priors from the public data for gradient matching approach. We need to clarify that their learning paradigms are totally different from ours. Nevertheless, We acknowledge that UNet is also a possible architecture of the attack model, the complementary experiments are shown in the table below (also in Table to in the revision).
> >
> > This table shows quantitative comparison of GIT with prior works on different networks and batch sizes. We use MLP \& UNet to represent the generative method using a fixed MLP \& UNet architecture, which shares similar number of parameters to our proposed method. The numbers in the table represent the MSE between the reconstructed data and the ground truth on the test set. We use $10000$ samples to train generative models.
> >
> > | **Leaked Model**        | **Method** | **Metrics** | **Batch Size = 1** | **Batch Size = 2** | **Batch Size = 4** |
> > |--------------------------|------------|-------------|--------------------|--------------------|--------------------|
> > | **LeNet (5 layers)**     | **DLG**    | **MSE**     | **0.0008**         | 0.0472             | 0.0975             |
> > |                          |            | **PSNR**    | **30.97**          | 13.26              | 10.11              |
> > |                          | **MLP**    | **MSE**     | 0.0241             | 0.0332             | 0.0571             |
> > |                          |            | **PSNR**    | 16.18              | 14.79              | 12.43              |
> > |                          | **GIT**    | **MSE**     | 0.0099             | **0.0122**         | **0.0254**         |
> > |                          |            | **PSNR**    | 20.04              | **19.14**          | **15.95**          |
> > |                          | **UNet**   | **MSE**     | 0.0316             | 0.0393             | 0.0435             |
> > |                          |            | **PSNR**    | 15.00              | 14.06              | 13.62              |
> > | **ResNet (20 layers)**   | **DLG**    | **MSE**     | 0.1202             | 0.1347             | 0.1365             |
> > |                          |            | **PSNR**    | 9.20               | 8.71               | 8.65               |
> > |                          | **MLP**    | **MSE**     | 0.0354             | 0.0473             | 0.0589             |
> > |                          |            | **PSNR**    | 14.51              | 13.25              | 12.30              |
> > |                          | **GIT**    | **MSE**     | **0.0193**         | **0.0246**         | **0.0388**         |
> > |                          |            | **PSNR**    | **17.14**          | **16.09**          | **14.11**          |
> > |                          | **UNet**   | **MSE**     | 0.0515             | 0.0560             | 0.0619             |
> > |                          |            | **PSNR**    | 12.88              | 12.52              | 12.02              |
> >
> > **References**
> >
> > [1] Jinwoo Jeon, Kangwook Lee, Sewoong Oh, Jungseul Ok, et al. Gradient inversion with generative image prior. Advances in neural information processing systems, 34:29898–29908, 2021.

---

> > > ### Comment · Reviewer_tY9t · 2024-11-26
> > > **Response during Discussion**
> > >
> > > My concerns of weakness in theoretical side has been addressed.
> > >
> > > I appreciate the authors efforts in providing results in an additional dataset.
> > >
> > > I raise the score to 5.
> > >
> > > Best,
> > >
> > > reviewer tY9t

---

### Official Review · Reviewer_TtUQ · 2024-11-03

**Soundness:** 2
**Presentation:** 2
**Contribution:** 2
**Rating:** 5
**Confidence:** 3

**Summary:**

This paper introduces GIT, a novel gradient reconstruction attack designed for scenarios where the parameters of the victim models are unknown and only partial training data is accessible. GIT first leverages the available training data to pre-train a proxy model that can generate gradients identical to those produced by the victim client models when given the same data. This is accomplished by computing the inverse of each layer and minimizing the distance between the final reconstructed input and the original data. Once trained, the proxy model enables the attacker to reconstruct the victims' private data upon receiving gradients from the clients. Experimental results demonstrate that GIT achieves lower mean squared error (MSE) compared to previous methods.

**Strengths:**

1. The task considered in this paper is meaningful.
2. The attempt to relax the assumption and find a proxy model is interesting.

**Weaknesses:**

1. My main concern is the assumption. While the authors claim that they relax assumptions, their assumptions are impractical or barely different from those of the existing works.
    - Access to training data. This violates the privacy protection attempt in FL in the first place. The authors currently use up to 10000 training images, i.e., 20% of the original dataset size. While people could argue that some publicly available data is shared online, the authors should design more coherent experiments to support their claim and conduct an ablation study on the effect of available data points.
    - The training data labels are unknown. The authors assume access to the training data but not labels, which is counterintuitive.
    - No access to clients' model parameters but per-client gradients. While the authors claim the former, their approach highly relies on the latter, which seems equivalent and thus contradicts their claim. A more reasonable problem formulation would be to use the aggregated gradients solely. This is more practical as the FL server can only observe the aggregated gradients when using protocols like homomorphic encryption.
2. Claims of offline training. Despite the claim, the authors require clients to produce gradients. How do they do it offline?
3. Limited experiments.
    - The authors conduct experiments only on two models and one dataset.
    - The considered baselines might not be valid as they consider totally different assumptions.
    - (Minor) While I understand it is challenging and may take time to solve, the authors only consider small batch sizes and small images.
4. Experiment design.
    - More insightful analysis could be conducted, such as the distance between the learned weights and the victim model weights and performance when using OOD or hold-out datasets.
    - Moreover, the authors currently only report MSE errors. It is known that MSE errors might not directly translate to visual quality. It would be interesting to additionally measure perception loss or inception score. Otherwise, as the results presented in Figures 2 and 3 show, it is difficult to judge what kind of information is leaked.
    - Can the proposed method scale up to larger images beyond cifar10?

Overall, while the proposed technique is interesting, it might not fit in the application or is not ready for publication at this point.

Editorial comments:
1. (minor) I recommend the authors provide an overview and state the contributions at the beginning of Sec. 3. Given the current presentation, I feel the readers might get lost.
2. The term "threat model" often refers to the problem settings and the assumptions for both the attacker and the defender in most privacy, security-related work.

**Questions:**

1. What does $\sigma^\prime$ mean in L167?
2. In L194, there are two $\sigma$.

---

> ### Author Response · Authors · 2024-11-23
>
> **Answer to Weakness 1. Access to Training Data**
>
> The same assumptions in our work have been investigated in existing literature as well (see [1]), where the attacker has access to some input-gradient pairs of local dataset. Our assumptions are based on the circumstance where some local dataset and its corresponding gradients on one client are leaked to the attacker, which is indeed different from training data reconstruction approach by gradient matching  (see [2], [3], [4] and [5]).
>
> The ablation study over the number of input-gradient pairs to train the generative model is conducted in Table 5 of Section 5.4 in revision.
>
> **Answer to Weakness 1. Training Data Labels**
>
> In most existing training data reconstruction approach by gradient matching, labels and back propagation process are necessary (see [2], [3], [4] and [5]). While generative method (see [1]) including our work does not rely on the labels of training dataset, although it may be leaked to the attacker along with the training data as well.
>
> **Answer to Weakness 1. Access to Gradients**
>
> Our proposed method is based on the gradient leakage during federated learning settings like Federated Averaging (FedAvg). In FedAvg, during a communication round, each client trains the model on its local dataset and then transmit the gradients to the central server to update the model. To the best of our knowledge, most existing works of training data reconstruction (see [2], [3], [4], [5] and [6]) also assume that the attacker is able to intercept gradients transmitted from client nodes to the server. Since the central server is not hacked by the attacker, we have no access to the model parameters there.
>
> **Answer to Weakness 2**
>
> Calculating and transmitting gradients are the essential processes of FedAvg. The attacker is able to intercept these gradients and we utilize them to train generative models for training data reconstruction. By contrast, gradient matching approaches require online access to the back propagation of the model on the central server during attack while generative approaches do not.
>
> **Answer to Weakness 3. Multiple Models and Datasets**
>
> The complementary experiments for additional dataset TinyImageNet-200 is shown in the table below. The reconstructed images are shown in  Appendix E.2 in revision, with its best MSE of 0.0317 $\pm$ 0.0003. The results demonstrate that high-frequency information, including object contours and background details, is effectively recovered. Although the MSE and visual quality are lower compared to CIFAR10 under the same configuration, reconstructing data from higher-resolution images poses a significant challenge. Notably, resolutions larger than CIFAR10 have not been explored in baseline methods MLP and DLG.
>
> | **Leaked Model**    | **Metrics** | **Batch Size = 1** | **Batch Size = 2** | **Batch Size = 4** |
> |----------------------|-------------|--------------------|--------------------|--------------------|
> | **LeNet 5**         | **MSE**     | 0.0317             | 0.0437             | 0.0509             |
> |                      | **PSNR**    | 14.99              | 13.60              | 12.93              |
> |----------------------|-------------|--------------------|--------------------|--------------------|
> | **ResNet 20**       | **MSE**     | 0.0983             | 0.1147             | 0.1274             |
> |                      | **PSNR**    | 10.07              | 9.40               | 8.95               |
>
> **Answer to Weakness 3. Different assumptions of baselines**
>
> There are two baselines considered in Section 5, i.e. MLP \& DLG. MLP represents the generative approach using a fixed MLP architecture, which has the same assumptions as ours. While gradient matching approaches have different assumptions, we conduct experiments on DLG, the most popular framework of gradient matching for a more comprehensive comparison and reference of existing training data reconstruction approaches.
>
> **Answer to Weakness 3. Batch Size \& Image Resolution**
>
> For larger image resolution, refer to **Answer to Weakness 3. Multiple Models and Datasets**. The information loss caused by large batch sizes is a critical issue commonly faced in training data reconstruction, which requires specific methodology to solve (see [5], [6], [7] and [8]). In addition, additional information is needed in these approaches focusing on solving large batch restoration. In this paper, we concentrate on achieving optimal recovery performance with the same model complexity under weaker assumptions by designing a robust and efficient attack model structure. Therefore, large batch size reconstruction is out of the scope of our work.

---

> > ### Author Response · Authors · 2024-11-23
> >
> > **Answer to Weakness 4. Distance between Weights**
> >
> > The complementary experiments on distance between weights are conducted on FineGIT, since the parameters of FineGIT are the estimation of leaked model's weights (As shown in Algorithm 1 in the paper), while the parameter of CoarseGIT is the black-box approximation.
> >
> > Section 5.5 in revision plots the curve of L2 distance between weights of the attack model and the leaked model, as well as the MSE between reconstructed input and the ground truth input. Figure 3 in the paper illustrates the L2 distance curve between the attack model's weights and the leaked model's weights, alongside the MSE between the reconstructed inputs and the ground truth inputs. As shown in the figure, when FineGIT converges, its weights align closely with the ground truth weights. This convergence highlights the effectiveness of FineGIT in extracting weight information from the leaked model.
> >
> > **Answer to Weakness 4. Metrics**
> >
> > The complementary experiments of additional metrics PSNR are shown in the table below (also in Table 2 in the paper). This table shows quantitative comparison of GIT with prior works on different networks and batch sizes. We use MLP \& UNet to represent the generative method using a fixed MLP \& UNet architecture, which shares similar number of parameters to our proposed method. The numbers in the table represent the MSE between the reconstructed data and the ground truth on the test set. We use $10000$ samples to train generative models.
> >
> > Results analysis: The results in Figure 2 and 5 (in Appendix E) in the paper illustrate that the low frequency information of the original images is more likely to lose during reconstruction because of convolution operation.
> >
> > | **Leaked Model**        | **Method** | **Metrics** | **Batch Size = 1** | **Batch Size = 2** | **Batch Size = 4** |
> > |--------------------------|------------|-------------|--------------------|--------------------|--------------------|
> > | **LeNet (5 layers)**     | **DLG**    | **MSE**     | **0.0008**         | 0.0472             | 0.0975             |
> > |                          |            | **PSNR**    | **30.97**          | 13.26              | 10.11              |
> > |                          | **MLP**    | **MSE**     | 0.0241             | 0.0332             | 0.0571             |
> > |                          |            | **PSNR**    | 16.18              | 14.79              | 12.43              |
> > |                          | **GIT**    | **MSE**     | 0.0099             | **0.0122**         | **0.0254**         |
> > |                          |            | **PSNR**    | 20.04              | **19.14**          | **15.95**          |
> > |                          | **UNet**   | **MSE**     | 0.0316             | 0.0393             | 0.0435             |
> > |                          |            | **PSNR**    | 15.00              | 14.06              | 13.62              |
> > | **ResNet (20 layers)**   | **DLG**    | **MSE**     | 0.1202             | 0.1347             | 0.1365             |
> > |                          |            | **PSNR**    | 9.20               | 8.71               | 8.65               |
> > |                          | **MLP**    | **MSE**     | 0.0354             | 0.0473             | 0.0589             |
> > |                          |            | **PSNR**    | 14.51              | 13.25              | 12.30              |
> > |                          | **GIT**    | **MSE**     | **0.0193**         | **0.0246**         | **0.0388**         |
> > |                          |            | **PSNR**    | **17.14**          | **16.09**          | **14.11**          |
> > |                          | **UNet**   | **MSE**     | 0.0515             | 0.0560             | 0.0619             |
> > |                          |            | **PSNR**    | 12.88              | 12.52              | 12.02              |
> >
> > **Answer to Weakness 4. Image Resolution**
> >
> > Refer to answer to **Answer to Weakness 3. Multiple Models and Datasets**.
> >
> > **Answer to Editorial Comments**
> >
> > We agree that changing "threat model" to "attack model" would better reflects its meaning in the given context.
> >
> > **Answer to Questions**
> >
> > $\sigma'$ refers to derivative of activation function $\sigma_i(z_i)$. Other modifications for editorial comments please refer to revision.

---

> > > ### Author Response · Authors · 2024-11-23
> > >
> > > **References**
> > >
> > > [1] Ruihan Wu, Xiangyu Chen, Chuan Guo, and Kilian Q Weinberger. Learning to invert: Simple adaptive attacks for gradient inversion in federated learning. In Uncertainty in Artificial Intelligence, pp. 2293–2303. PMLR, 2023.
> > >
> > > [2] Ligeng Zhu, Zhijian Liu, and Song Han. Deep leakage from gradients. In H. Wallach, H. Larochelle, A. Beygelzimer, F. d'Alch´e-Buc, E. Fox, and R. Garnett (eds.), Advances in Neural Information Processing Systems, volume 32. Curran Associates, Inc., 2019.
> > >
> > > [3] Bo Zhao, Konda Reddy Mopuri, and Hakan Bilen. idlg: Improved deep leakage from gradients. arXiv preprint arXiv:2001.02610, 2020.
> > >
> > > [4] Jonas Geiping, Hartmut Bauermeister, Hannah Dr¨oge, and Michael Moeller. Inverting gradients-how easy is it to break privacy in federated learning? Advances in neural information processing systems, 33:16937–16947, 2020.
> > >
> > > [5] Hongxu Yin, Arun Mallya, Arash Vahdat, Jose M Alvarez, Jan Kautz, and Pavlo Molchanov. See through gradients: Image batch recovery via gradinversion. In Proceedings of the IEEE/CVF conference on computer vision and pattern recognition, pp. 16337–16346, 2021.
> > >
> > > [6] Junyi Zhu and Matthew Blaschko. R-gap: Recursive gradient attack on privacy. arXiv preprint
> > > arXiv:2010.07733, 2020.
> > >
> > > [7] Kailang Ma, Yu Sun, Jian Cui, Dawei Li, Zhenyu Guan, and Jianwei Liu. Instance-wise batch label restoration via gradients in federated learning. In The Eleventh International Conference on Learning Representations, 2023.
> > >
> > > [8] Dimitrov et al. "SPEAR: Exact Gradient Inversion of Batches in Federated Learning", https://arxiv.org/abs/2403.03945

---

> ### Comment · Reviewer_TtUQ · 2024-11-26
> **Response to the rebuttal**
>
> I want to thank the authors for the detailed response. After carefully reading the rebuttal, my concern remains, and I'd like to comment on the following points.
>
> - **Access to training data and labels**
>
> As discussed in my initial comment, I agree that the scenario might exist. My concern, beyond availability, is the experiment design. The effect and practical implications of leaked images are unclear compared to existing works that do not require any available images. Specifically, (1) When does an attack have access to up to 20% (10000 out of 50000 for CIFAR-10) of training images in practice? (2) Why 20%, not other numbers? (3) What if only fewer images, say 10% and 5%, are available? (4) The author use small batch size in most of the experiments. How do the authors report the performance? Do they measure the performance on images that are available to attackers and hold-out images (namely, the other images that are unseen to the attacker) separately? As these aspects are missing in the paper, I am not fully convinced by the setting.
>
> - **Parameters are unkown**
> I found the authors' point unclear. In federated learning, all clients typically start with the same initialization. This means that if an attacker knows the gradients, they can inherently deduce the model parameters.
>
> - **Claims of offline training.**
> My understanding of offline training is that the generative model can be pre-trained before initiating the federated learning process and querying the clients. However, the proposed method appears to require collecting gradients during the FL training process, which, in my view, does not qualify as offline training.
>
> - **Multiple Models and Datasets**
> While it is great to see that the proposed method can be applied to higher-resolution images, the poor visual quality (on both CIFAR-10 and TinyImageNet) does not appear to leak information.
>
> - **Distance between Weights**
> Thanks for the insight. Given that GIT is an approximation approach, it is surprising that both models will be almost identical in the end.
> Could the authors briefly clarify the scenarios in which fineGIT is most suitable and those where coarseGIT would be more appropriate, particularly regarding the assumption?
>
> Overall, while the authors' response addressed some of my concerns, my biggest concern remains in the setting. Given the current presentation, I'd lean toward keeping my current score but stay open to discussion.

---

> > ### Author Response · Authors · 2024-12-03
> >
> > **Answer to: Access to training data and labels**
> >
> > (1) In the existing work (see [1]), the authors randomly subsample the CIFAR-10 training set to construct leaked datasets of various sizes: {500, 5000, 15000, 25000, 35000, 45000, 50000}, including using the entire CIFAR-10 dataset to evaluate the performance of their methods. Therefore, we use 10,000 data instances in most evaluations to facilitate comparison with baselines. **Nevertheless, we conducted ablation studies on the size of data volume in Table 4 and Figure 2 and demonstrated that our proposed GIT performs better than baselines and can successfully reconstruct input data with as few as 1,000 data points.** Additionally, it is worth noting that if model querying is allowed (a setting widely used in gradient inversion attacks, see [2], [3], [4], and [5]), public data can be used to generate input-gradient pairs, eliminating the need to obtain real leaked data.
> >
> > (2) We selected 10,000 data points (20\% of the CIFAR-10 training set) because baseline generative methods using MLP require a minimum of 5,000 data points to outperform other gradient inversion methods. To ensure sufficient data for effective attacks, we chose twice that amount. Subsequently, we supplemented the experiments by considering smaller data volumes ({1,000, 5,000, and 10,000}) for comparison in Table 4 and Figure 2.
> >
> > (3) Results for data volumes corresponding to 20\%, 10\%, and 2\% of the CIFAR-10 training set are presented in Table 4 and Figure 2 in the Training Data Volume section.
> >
> > (4) We used certain proportions (20\%, 10\%, and 2\%) of the CIFAR-10 training set as the leaked data to train the attack model. The CIFAR-10 test set was used as the evaluation dataset to measure the attack model's performance in reconstructing data. **Therefore, the performance we report is based on datasets that are unseen by the attacker but share the same distribution as the leaked data.**
> >
> > **Answer to: Parameters are unkown**
> >
> > In our settings, we assume the attacker has access to only some, but not all, clients in FL. In this context, the attacker has no access to the model parameters of the clients that are not hacked. The attacker only has the input-gradient pairs from the hacked clients and the shared gradients from the other clients. Although the model parameters are synchronised from time to time (not necessarily synchronised after each update iteration), the attacker has no access to model parameters on which the leaked gradients are calculated.
> >
> > **Answer to: Claims of offline training**
> >
> > For generative methods to conduct gradient inversion, offline training is necessary. This is also true for other existing generative methods. We believe this setting is practical, because after hacking the clients of an FL system, the attacker already has several input-gradient pairs; it **does not need to wait for the gradient exchange from other clients to start training the generative model**.
> >
> > In addition, we would like to point out that although generative methods need offline training, it is **very efficient in the inference phase**, because it does not involve iterative optimizations during the inference phase by querying the model, like in DLG methods.
> >
> > **Answer to: Distance between Weights**
> >
> > The parameter to be trained in FineGIT is an estimation of the parameters of the model under attack, and thus has the same dimensionality. According to our recursive inverse equation (see Equation (4)), the goal of training is to recover the parameters on which the leaked gradients are calculated, so the parameters of FineGIT will gradually converge to the ground truth parameters, demonstrating its effectiveness. However, as the model depth increases (specifically for networks with more than 5 layers in practice), the instability caused by the computation of the pseudo-inverse leads to a significant degradation in performance. To address this, we use CoarseGIT as an approximation. Unlike FineGIT, CoarseGIT estimates the mapping in a black-box manner, meaning its parameters may not match the dimensionality of the model under attack. In practice, coarseGIT is applied in networks with more than 5 layers or smaller kernel sizes (which result in fewer information being available in the leaked gradients).

---

### Meta-Review · Area_Chair_CX3e · 2024-12-26

**Metareview:**

The paper proposes the Gradient Inversion Transcript (GIT), a method for reconstructing training data in federated learning by leveraging gradient information rather than model parameters. The approach aims to work in scenarios where only gradients are shared, offering a theoretically grounded alternative to existing gradient inversion techniques. While GIT introduces an adaptive generative model tailored to the target architecture, showing some empirical advantages, but  the reviewers found that the paper’s theoretical contributions are limited, relying on a straightforward extension of gradient backpropagation. The justification for using certain approximations (like the pseudo-inverse) is unclear. There was a lack of diversity in the experiments and comparison with a broader set of baselines, which weakens the claims of superiority. The paper also make  unrealistic assumptions, and while acknowledges overfitting but does not offer solutions, and it remains unclear whether GIT can scale to larger datasets or more complex models. Due to these issues, the consensus was to reject the paper.

**Additional Comments On Reviewer Discussion:**

The proposed method is an interesting and novel approach, but it requires significant refinement in terms of both theoretical justification and empirical validation before it can be considered for publication including better clarity on assumptions and threat model, expanding empirical results by considering diverse architectures and datasets, and improving the theoretical justification.

---

### Decision · Program_Chairs · 2025-01-22

Reject